# Diversity amongst human cortical pyramidal neurons revealed via their sag currents and frequency preferences

Homeira Moradi Chameh [1], Scott Rich [1], Lihua Wang[1], Fu-Der Chen[2,3], Liang Zhang[1,4], Peter L. Carlen [1,4,5], Shreejoy J. Tripathy[6,7,8,10] & Taufik A. Valiante [1,2,5,7,9,10 ✉]

In the human neocortex coherent interlaminar theta oscillations are driven by deep cortical layers, suggesting neurons in these layers exhibit distinct electrophysiological properties. To characterize this potential distinctiveness, we use in vitro whole-cell recordings from cortical layers 2 and 3 (L2&3), layer 3c (L3c) and layer 5 (L5) of the human cortex. Across all layers we observe notable heterogeneity, indicating human cortical pyramidal neurons are an electrophysiologically diverse population. L5 pyramidal cells are the most excitable of these neurons and exhibit the most prominent sag current (abolished by blockade of the hyperpolarization activated cation current, $I_h$). While subthreshold resonance is more common in L3c and L5, we rarely observe this resonance at frequencies greater than 2 Hz. However, the frequency dependent gain of L5 neurons reveals they are most adept at tracking both delta and theta frequency inputs, a unique feature that may indirectly be important for the generation of cortical theta oscillations.

[1] Krembil Brain Institute, University Health Network, Toronto, ON, Canada. [2] Department of Electrical and Computer Engineering, University of Toronto, Toronto, ON, Canada. [3] Max Planck Institute of Microstructure Physics, Halle, Germany. [4] Departments of Medicine & Physiology, University of Toronto, Toronto, ON, Canada. [5] Institute of Biomedical Engineering, University of Toronto, Toronto, ON, Canada. [6] Krembil Centre for Neuroinformatics, Centre for Addiction and Mental Health, Toronto, ON, Canada. [7] Institute of Medical Sciences, University of Toronto, Toronto, ON, Canada. [8] Department of Psychiatry, University of Toronto, Toronto, ON, Canada. [9] Division of Neurosurgery, Department of Surgery, University of Toronto, Toronto, ON, Canada. [10] These authors contributed equally: Shreejoy J. Tripathy, Taufik A. Valiante. ✉email: taufik.valiante@uhn.ca

Comparative studies between human and rodent cortical neuronal physiology have revealed unique human cortical neuronal and microcircuit properties. At the cellular level, human neurons have been shown to have unique morphological properties[1], potentially reduced membrane capacitances[2], increased dendritic compartmentalization in thick-tufted L5 pyramidal cells[3], higher h-channel densities in L3 versus L2 pyramidal cells[4], and a wholly unique neuronal cell type[5,6]. At the microcircuit level, human neocortical circuits demonstrate unique reverberant activity[7], different spike-timing-dependent plasticity rules compared to neocortical circuits in rodents[8], and coherent oscillations between superficial and deep cortical layers[9]. In addition, correlations between patient intelligence quotient and cellular features of human layer 2, 3, and 4 pyramidal cells have been demonstrated in both action potential (AP) kinetics and the length and complexity of dendritic arbors[10].

Although understanding the unique biophysical and synaptic properties of neurons experimentally remains an important endeavor, computational models and mathematical formulations of neurons and circuits are essential for describing and explaining mesoscopic-level collective dynamics, such as oscillations[11–13]. Indeed, it has been recently posited that "a set of brain simulators based on neuron models at different levels of biological detail" are needed in order to "allow for systematic refinement of candidate network models by comparison with experiments"[14]. By extension, to create simulations of the human brain and cortical microcircuit, we need neuronal models derived from direct human experiments. Thus, as we explore what is uniquely human about the human brain in order to, for example, tackle the increasing societal burden of neurological and neuropsychiatric conditions[15,16], infusing computational models with human derived microscopic and mesoscopic cellular and circuit properties will be critically important.

In this context, our previous experiments in human cortical slices have demonstrated that spontaneous theta-like activity, the most ubiquitous oscillation in the human brain[17], can be induced by application of cholinergic and glutamatergic agonists[9]. We observed theta oscillations that were coherent between cortical laminae, with the deep layer leading in phase relative to the superficial layer[9]. We also observed robust cross-frequency coupling between theta and high-gamma activity that was modulated with the strength of synchrony between cortical laminae[18]—so called coordination though coherent phase–amplitude coupling[19]. Given the role of intrinsic electrophysiological properties in the generation of oscillations[13] and the finding that deep layer theta leads superficial layer theta in phase, we reasoned that deep layer neurons in the human neocortex are likely endowed with distinct biophysical properties that enable them to "drive" such interlaminar activity.

One of the candidate membrane currents thought to contribute to low-frequency (<8 Hz) oscillations is the hyperpolarization activated cation current or h-current ($I_h$)[3,4,20,21]. This current is important for oscillations and pacemaking activity in a myriad of cell types, ranging from midbrain and hippocampal neurons to cardiac pacemaker neurons[22–24]. Consistent with its role in contributing to resonant activity, a recent study in the human neocortex demonstrated that $I_h$ appeared necessary for the subthreshold resonance observed in L3 neurons[4]. In addition, it has been reported that thick-tufted neurons in L5 of the human neocortex also display prominent somatic and dendritic $I_h$ and subthreshold resonance[3]. However, recent transcriptomic evidence and detailed comparisons to homologous cells in rodents[25] have suggested that these thick-tufted, extratelencephalic (ET) neurons are much rarer, implying that our understanding of the electrophysiological properties of L5 pyramidal neurons remains incomplete.

Based on our previous findings that deep layer activity appears to drive superficial activity in the human cortex[9], we hypothesized that this "leading" role in generating interlaminar coherence can be attributed in part to the differing intrinsic properties of deep layer from superficial layer neurons. In pursuit of this hypothesis, we sought to gain a more complete understanding of the features of human L5 cortical pyramidal neurons. We used whole-cell recordings to characterize pyramidal cells in L2&3, L3c, and L5, focusing on the amplitude and kinetics of $I_h$ via the sag voltage. In addition to key biophysical differences favoring greater excitability in human L5 versus L2&3 pyramidal cells, we found that L5 and L3c demonstrated larger sag voltage amplitudes relative to L2&3 pyramidal cells generally. Somewhat surprisingly, while some subthreshold resonance at >2 Hz was observed in our experiments, we found this feature to be generally quite rare among pyramidal cells across all layers. However, we did find that L5 pyramidal cells showed enhanced frequency-dependent gain at delta and theta frequencies, which motivates our presentation of a "dynamic circuit motif" (DCM)[13] underlying how L5 neurons "drive" human cortical theta. Lastly, we found notable cell-to-cell variability in electrophysiological parameters sampled from pyramidal cells recorded within the same lamina consistent with previous studies in human L2&3[4,26,27] and further reveal that this variability is especially large in L5.

## Results

Whole-cell patch clamp recordings were obtained from human neocortical neurons located in L2&3, L3c, and L5 within acute brain slices collected from 61 patients. Tissues were obtained primarily from patients who underwent resective surgery for pharmacologically intractable epilepsy (see Table 1 for a summary of patient details). For many of our recordings, we did not annotate our L2&3 pyramidal cells as specifically belonging to either L2 or L3 considering that the majority of these data were collected prior to publication of a recent paper illustrating divergent electrophysiological and morphological features of these neurons in the human neocortex[4]. Consequently, we later recorded from a targeted set of pyramidal cells in L3c (i.e., the deepest part of L3) to specifically contrast and compare our findings with those from previous findings[4].

**Diverse morphologies and passive membrane properties of pyramidal cells in L2&3, L3c, and L5.** To confirm the successful targeting of pyramidal cells, a subset of neurons was filled with biocytin and underwent subsequent morphological reconstruction. Figure 1a shows example electrophysiological sweeps of L2&3, L3c, and L5 pyramidal cells with corresponding three-dimensional (3D) morphological reconstructions.

The 3D reconstructions revealed a rich diversity of human pyramidal cell morphologies, consistent with recent detailed demonstrations of the distinct cellular morphologies of human cortical neurons as a function of cortical lamina[1,3,4,26]. Pyramidal cells with somas located in L2 and the upper part of L3 had complex basal dendrites, with apical dendrites often reaching L1. Pyramidal cells located in L3c showed different morphologies, with one cell (cell d) showing simple basal dendrites and another (cell e) showing much more complex basal dendrites, consistent with recent reports on the heterogeneity of pyramidal cells in L3[26]. Lastly, we observed two L5 neurons with very different morphologies: one cell (cell f) displays a simple morphology with apical dendrites terminating at the border of L3 and L4, and another, considerably larger pyramidal cell (cell g) with a highly complex basal dendrite and two apical dendrite trunks, with one trunk terminating in upper L3 and the other projecting to lower

**Table 1 Demographic data (for a subset of 49 patients where such information was available).**

| Age (years) | Sex | Years of seizure history | Diagnosis | Antiepileptic drugs | Resection location |
|---|---|---|---|---|---|
| 39 | F | 11 | Tumor | LSC, LRZ, LEV | Right ATL |
| 58 | F | 8 | Tumor | CBZ | Left FL |
| 57 | M | 45 | Epilepsy | LSC, CZP, CBZ | Right ATL |
| 27 | M | 11 | Epilepsy | LSC, LRZ, CLB | Right ATL |
| 24 | M | 8 | Epilepsy | LEV, LTG | Right ATL |
| 25 | M | 12 | Epilepsy | CBZ, LSC | Right ATL |
| 33 | F | 4 | Epilepsy | LEV | Left FL |
| 33 | M | 14 | Epilepsy | PHN, LEV | Right FL |
| 22 | M | 6 | Epilepsy | PHN, CBZ, LTG | Left ATL |
| 21 | M | 2 | Epilepsy | DR, CLB, MJ | Left parietal lobe |
| 22 | M | 12 | Epilepsy | PHN, LRZ | Right ATL |
| 23 | F | 23 | Epilepsy | CBZ, LEV, LSC | Right FL |
| 53 | F | 43 | Epilepsy | CBZ, LSC, LEV | Left ATL |
| 37 | F | 2 | Tumor | GPN, LSC, LEV, CLB, LRZ | Right FL |
| 47 | F | 4 | Epilepsy | CBZ, CLB | Left ATL |
| 52 | M | 13 | Epilepsy | CBZ, CLB | Left ATL |
| 50 | F | 26 | Epilepsy | PHN, LTG | Right ATL |
| 36 | F | 34 | Epilepsy | LSC, CBZ | Left ATL |
| 40 | M | 29 | Epilepsy | LEV | Right ATL |
| 25 | F | 10 | Epilepsy | CBZ, LSC, LEV | Right ATL |
| 52 | M | 27 | Epilepsy | LSC, LRZ | Left ATL |
| 21 | M | 11 | Epilepsy | LTG, CBZ | Left ATL |
| 63 | M | 0.1 | Tumor | PHN | Right parietal lobe |
| 42 | M | 22 | Epilepsy | CBZ | Right FL |
| 25 | F | 22 | Epilepsy | LSC, CLB, LTG | Right FL |
| 24 | F | 3 | Tumor | LEV | Left ATL |
| 53 | M | 9 | Epilepsy | LEV | Left ATL |
| 45 | F | 20 | Epilepsy | LTG | Right ATL |
| 26 | F | 25 | Epilepsy | CBZ, CLB, LTG | Right ATL |
| 35 | F | 14 | Epilepsy | LRZ, DR, PHN | Left ATL |
| 24 | M | 6 | Epilepsy | LSC, LRZ, MJ | Right ATL |
| 53 | F | 51 | Epilepsy | LSC, CLB | Left ATL |
| 44 | F | 3 | Epilepsy | LTG | Left ATL |
| 25 | M | 14 | Epilepsy | CBZ | Right ATL |
| 19 | F | 15 | Epilepsy | PB, CLB, GPN, RFM | Right ATL |
| 30 | M | 12 | Epilepsy | PHN | Left ATL |
| 26 | M | 5 | Epilepsy | CBZ, DR | Right ATL |
| 28 | M | 13 | Epilepsy | CLB, MJ | Left ATL |
| 52 | F | 6 | Epilepsy | LTG, LEF | Left ATL |
| 26 | F | 9 | Epilepsy | ESL, TMP, CLB | Right ATL |
| 59 | F | 39 | Epilepsy | CLB, LSC | Right ATL |
| 37 | M | 5 | Epilepsy | CLB, LSC | Right ATL |
| 55 | M | 27 | Epilepsy | CLB, ESL, LTG | Left ATL |
| 42 | F | 3 | Epilepsy | CBZ, GPN | Right-ATL |
| 57 | F | 56 | Epilepsy | CLB, LTG, PRM, CBD oil | Left ATL |
| 24 | M | 4 | Epilepsy | LEV, LTG | Left ATL |
| 33 | M | 6 | Epilepsy | LSC, PGBPHN | Right ATL |
| 39 | M | 12 | Epilepsy | CLB, LTG, MJ | Left ATL |
| 36 | F | 16 | Epilepsy | LSC, PGB, PHN | Right ATL |

*CBZ* Carbamazepine, *CLB* Clobazam, *CZP* Clonazepam, *DR* Divalproex, *GPN* Gabapentin, *LEV* Levetiracetam, *LRZ* Lorazepam, *LSC* Lacosamide, *LTG* Lamotrigine, *MJ* Marijuana, *PB* Phenobarbital, *PHN* Phenytoin, *RFM* Rufinamide, *LEF* Leflunomide, *ESL* Eslicarbazepine acetate, *TMP* Tetramethylpyrazine, *PRM* primidone, *CBD* cannabidiol, *PGB* Pregabalin, *FL* frontal lobe.

L3 prior to its abrupt termination due to slicing or optical truncation.

While we note that our human L5 morphologies are different from those reported by Beaulieu-Laroche et al.[3] that targeted rare thick-tufted L5 pyramidal cells[25] with tufts reaching into L1[28], our cell morphologies are consistent with other previous reports that relatively few L5 neurons have dendrites extending past L3[1]. Additionally, given the challenge of potential dendrite truncation when preparing slices containing such large cells[1], it is possible that our representative L5 morphologies in Fig. 1 have been inadvertently truncated[1]. However, we only observed visible truncation in one branch of one cell (the largest cell, cell g) and no obvious truncation in the other cells shown in Fig. 1.

We next assessed the passive membrane properties (i.e., resting membrane potential (RMP), input resistance, and membrane time constant) of human cortical pyramidal cells in L2&3, L3c, and L5 ($n = 56$, $n = 15$, and $n = 105$ neurons, respectively) using hyperpolarizing current steps in current-clamp mode (see "Methods"). We found that passive membrane properties differed significantly between pyramidal cells of L2&3, L3c, and L5. Additionally, we found that L3c and L5 neurons had more depolarized RMPs relative to L2&3 neurons (L2&3: $-68.2 \pm 5.3$ mV, L3c: $-65.6 \pm 3.8$ mV, L5: $-65.6 \pm 6.5$ mV; Fig. 1b), with L5 neurons being significantly more depolarized at rest compared to L2&3 neurons ($p = 0.007$). We also found that L5 pyramidal cells showed higher input resistances relative to L2&3 and L3c

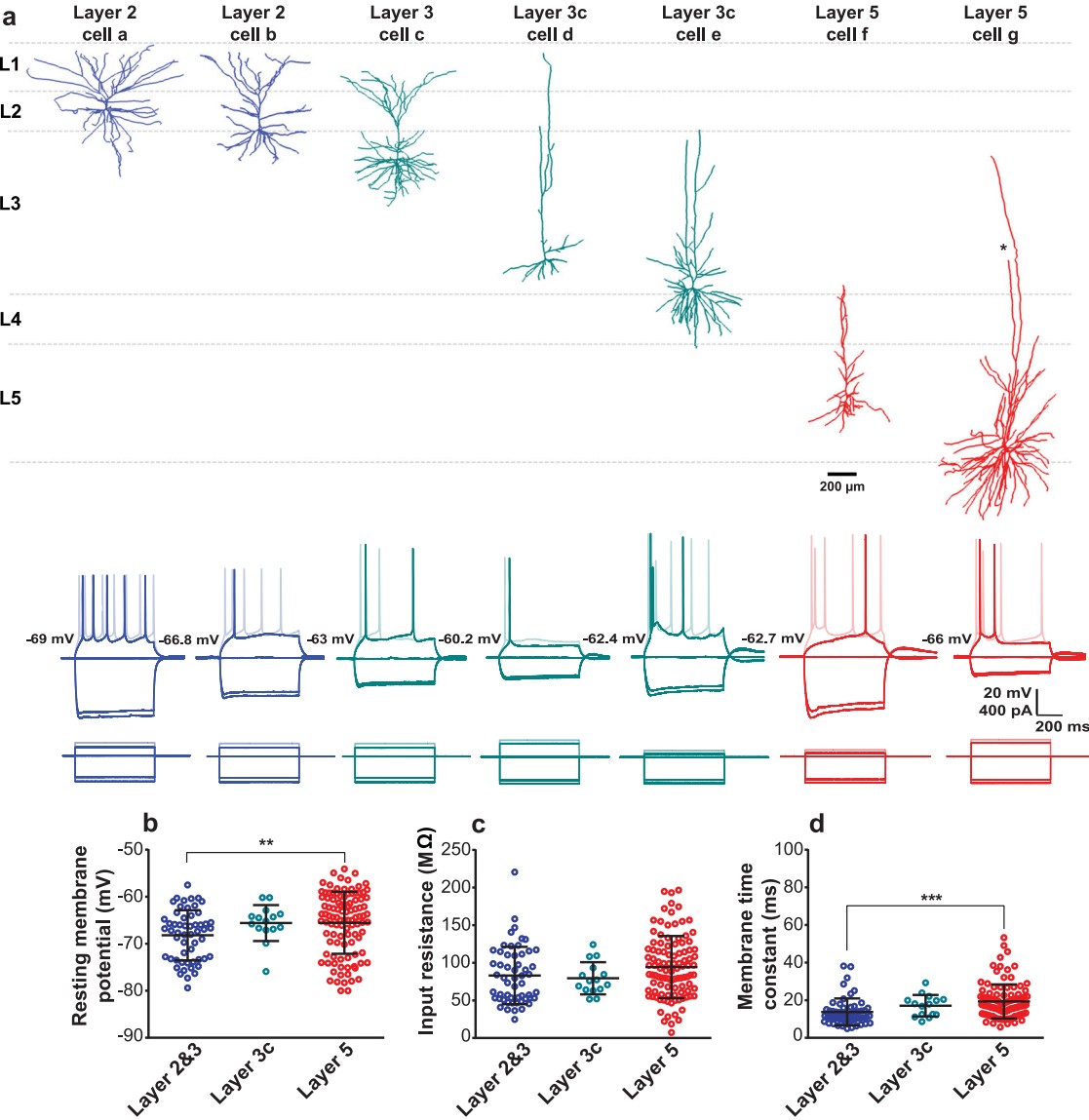

**Fig. 1 Diverse morphologies and passive membrane properties among pyramidal cells in the human neocortex. a** Example 3D reconstructions (top) and voltage traces (bottom) for L2&3, L3c, and L5 pyramidal cells following hyperpolarizing and depolarizing current injection. Cortical layer and relative position from pial surface are annotated for each reconstructed cell. Asterisk in one branch of apical dendrite in cell g with truncation (dendrite morphologies were otherwise not visibly truncated). **b–d** Resting membrane potentials ($p = 0.007$) (**b**), input resistances ($p = 0.111$) (**c**), and membrane time constants ($p < 0.0001$) (**d**) for pyramidal cells in L2&3, L3c, and L5. Error bars in **b–d** denote mean and standard deviations (SD). One-way ANOVA post hoc with Dunn's multiple comparison test were used for statistical comparison. L2&3 ($n = 56$), L3c ($n = 15$), and L5 ($n = 105$). ** denotes $p = 0.007$ and *** denotes $p < 0.001$. Source data are provided as a Source data file.

neurons (L2&3: $83 \pm 38.1$, L3c:$79.4 \pm 21.4$, L5: $94.2 \pm 41.3$ MΩ; Fig. 1c). This difference was not significant between layers ($p = 0.110$). L5 and L3c pyramidal cells also had slower membrane time constants ($\tau_m$) compared to L2&3 (L2&3: $13.7 \pm 7.1$, L3c: $17.1 \pm 5.7$, L5: $19.3 \pm 9.1$ ms, $p < 0.0001$; Fig. 1d).

In general, we found considerable electrophysiological heterogeneity among neurons sampled within each cortical layer, broadly consistent with the morphological reconstructions shown in Fig. 1a. For example, we found that pyramidal cells in L5 had input resistances as low as 20 MΩ and as high as 200 MΩ, possibly reflecting the dichotomy between thin- and thick-tufted pyramidal cells and/or the graded variation between pyramidal cells of varying sizes and dendritic complexities (as well as the potential inadvertent cutting of dendrites during slice preparation, see "Discussion").

We further compared these findings to published and publicly available datasets from human pyramidal cells. We note that the average input resistance among our population of recorded L5 pyramidal cells is considerably higher than that reported in Beaulieu-Laroche et al.[3], most likely due to differences in the neurons targeted for recordings between our studies. We also made use of a publicly accessible dataset of 272 pyramidal cells sampled from L2, L3, and L5 from an additional cohort of 39 human surgical patients characterized by the Allen Institute for Brain Sciences (http://celltypes.brain-map.org/). We note that, while the overall experimental design of the Allen Institute's dataset is similar to ours, there are some methodological differences, such as the composition of solutions used for slice preparation and recording (see "Methods"). The Allen Institute data are generally consistent with our finding that input

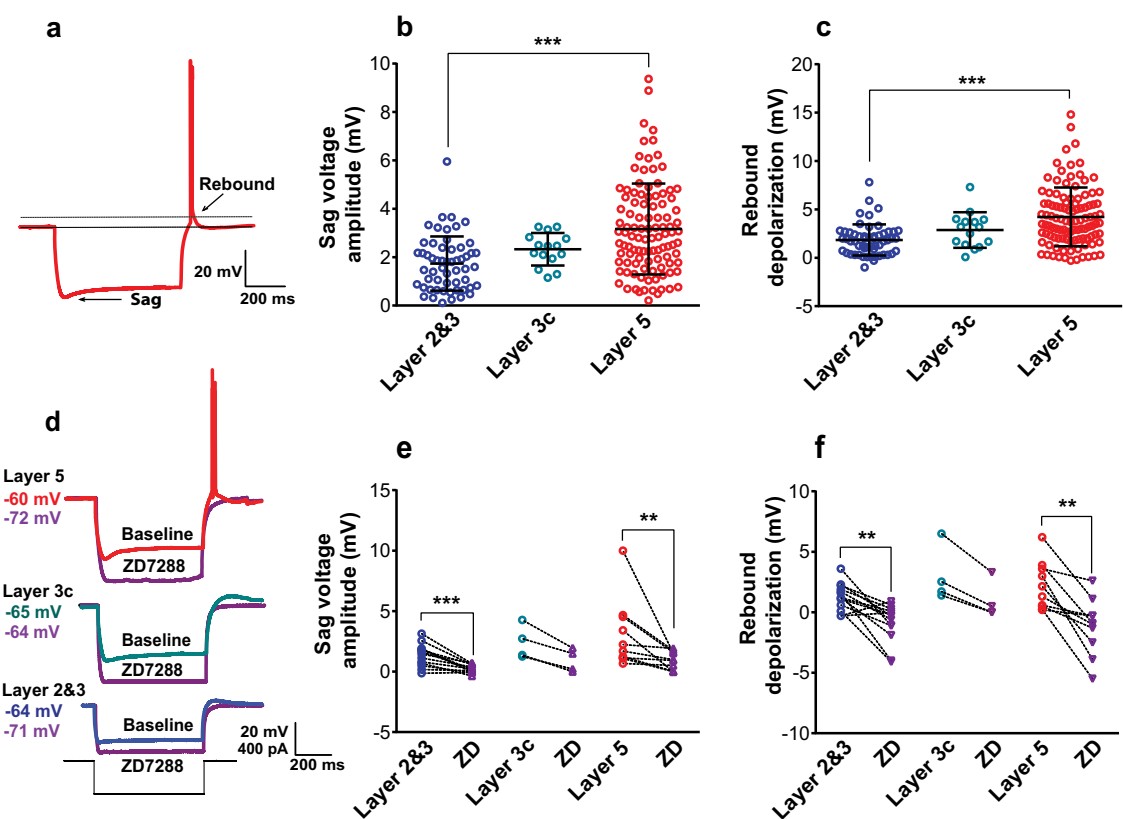

**Fig. 2 $I_h$-related membrane properties are more apparent in L5 pyramidal cells compared to L2&3 and L3c. a** Example voltage sweep from a representative L5 pyramidal cell during injection of −400 pA hyperpolarizing current step. Arrows indicate sag voltage and rebound depolarization and post-hyperpolarization rebound spiking. **b, c** Sag voltage amplitude ($p < 0.0001$; one-way ANOVA post hoc with Dunn's multiple comparison test, L2&3 ($n = 56$), L3c ($n = 15$), and L5 ($n = 105$)) (**b**) and rebound depolarization amplitude ($p < 0.0001$; one-way ANOVA post hoc with Dunn's multiple comparison test, L2&3 ($n = 56$), L3c ($n = 15$), and L5 ($n = 105$)) (**c**) among sampled L2&3, L3c, and L5 pyramidal cells in response to injection of hyperpolarizing current. Lines and error bars denote mean and SD and asterisks indicate significance of group comparison. **d** Example of voltage sweeps at baseline (red, green, and blue) and after (purple) bath application of the $I_h$ blocker ZD7288 (10 μM) following injection of −400 pA. **e, f** Bath application of ZD7288 diminished sag voltage amplitude (L2&3-ZD: $p = 0.001$, $n = 13$; L3c-ZD: $p = 0.125$, $n = 4$; L5-ZD: $p = 0.002$, $n = 10$; Wilcoxon matched-pairs signed rank test) (**e**) and rebound depolarization amplitude (L2&3-ZD: $p = 0.002$, $n = 13$; L3c-ZD: $p = 0.125$, $n = 4$; L5-ZD: $p = 0.002$, $n = 10$; Wilcoxon matched-pairs signed rank test) (**f**) in all layers. *** denotes $p \leq 0.001$ and ** denotes $p = 0.002$. Source data are provided as a Source data file.

resistances in L5 pyramidal cells are not smaller than those sampled in human L2 and L3 (Supplementary Fig. 1a) and that this trend holds even in neurons where the primary dendrites are not visibly truncated (Supplementary Fig. 1b). We further note that, while it appears L5 neurons have increased variability in these intrinsic properties relative to L2&3, the levels of heterogeneity are consistent with prior reports from L2&3 in previous human studies[4,26,27].

**Subthreshold active membrane properties of pyramidal cells in L2&3, L3c, and L5.** To assay sag voltage and rebound depolarization, we injected a series of hyperpolarizing currents (L2&3: $n = 56$, L3c: $n = 15$, L5: $n = 105$). L5 pyramidal cells had significantly larger sag voltage amplitudes than L2&3 pyramidal cells (L2&3: $1.7 \pm 1.1$ mV, L3c: $2.3 \pm 0.7$ mV, L5: $3.2 \pm 1.9$ mV, $p < 0.0001$ between L2&3 and L5; Fig. 2a, b). We found similar results using the dimensionless sag ratio measure that normalizes for input resistance differences between neurons (Supplementary Fig. 2a). We note that sag ratio is positively correlated with sag amplitudes (Supplementary Fig. 3, $r = 0.68$). These results were further replicated from the Allen Institute dataset (Supplementary Fig. 4; L2: $0.51 \pm 0.045$; L3: $0.125 \pm 0.067$; L5: $0.149 \pm 0.072$; $p = 2.21 \times 10^{-6}$ between L2 and L3, $p = 0.012$ between L3 and L5). These findings support recent evidence for a positive

correlation between sag voltage amplitude and distance from pial surface[4], with our results further extending this relationship to L5.

In addition, 21.9% of L5 neurons exhibited rebound spiking following the termination of a hyperpolarizing current pulse, whereas 1.8% of L2&3 neurons exhibited rebound spiking, and rebound spiking was not observed in L3c neurons. The rebound depolarization amplitude was significantly larger in L5 pyramidal cells compared to L2&3 and L3c neurons (L2&3: $1.8 \pm 1.6$ mV, L3c: $2.9 \pm 1.8$ mV, L5: $4.2 \pm 3$ mV, $p < 0.0001$ between L2&3 and L5; Fig. 2c).

To further characterize the $I_h$-specific component of membrane sag voltage, we bath applied the specific $I_h$ blocker ZD7288 (10 μm; ZD), with example traces shown in Fig. 2d. For L2&3 ($n = 13$), L3c ($n = 4$), and L5 ($n = 10$) pyramidal cells, after bath applying ZD we observed a significant reduction in voltage sag amplitude (L2&3: before $1.3 \pm 0.9$ mV, after $0.2 \pm 0.3$ mV, $p = 0.001$, L3c: before $2.4 \pm 1.4$ mV, after $0.9 \pm 0.9$ mV, $p = 0.125$, L5: before $3.1 \pm 2.8$ mV, after $0.9 \pm 0.8$ mV, $p = 0.002$; Fig. 2e) and in sag ratio (Supplementary Fig. 2b). Bath applying ZD7288 also significantly reduced the rebound depolarization amplitude (L2&3: before $1.3 \pm 1.1$ mV, after $-0.8 \pm 1.5$ mV, $p = 0.002$, L3c: before $3 \pm 2.3$ mV, after $0.9 \pm 1.6$ mV, $p = 0.125$, L5: before $2.1 \pm 1.9$ mV, after $-1.2 \pm 2.3$ mV, $p = 0.002$; Fig. 2f).

Voltage-clamp experiments were performed in a subset of neurons (L2&3: $n = 6$, L5: $n = 10$) to determine whether the $I_h$

amplitude differences arose from differences in channel kinetics between these two cell types. While space-clamp issues limit our ability to adequately voltage clamp distal cellular processes[29], we nevertheless considered it beneficial to use this technique to obtain semi-quantitative estimates of the amplitudes, kinetics, and voltage dependence of $I_h$ in human pyramidal cells. We used pharmacological blockers to specifically isolate $I_h$ (see "Methods"). We found that injecting voltage steps from −60 to −140 mV produced a slowly activating inward current (example traces shown in Supplementary Fig. 5a). Consistent with our current-clamp results, we found that the amplitudes of the $I_h$ were significantly smaller in L2&3 neurons compared with L5 (Supplementary Fig. 5c), whereas the time course of $I_h$ activation and the voltage sensitivity of $I_h$ (quantified at the half maximal activation voltage) was similar between L2&3 and L5 neurons (Supplementary Fig. 5b, d). These results suggest that the relatively larger L5 sag amplitude arises from increased channel numbers rather than differences in channel kinetics.

### Suprathreshold active membrane properties of pyramidal cells in layers 2/3, L3c, and 5.

Active membrane property differences between layers were characterized by examining the firing patterns of L2&3, L3c, and L5 human pyramidal cells ($n = 55$, $n = 15$, $n = 104$ neurons, respectively) using a series of depolarizing current pulses (0–400 pA, 600 ms) with examples shown in Fig. 3a–c. L5 neurons had a significantly larger AP than L2&3 neurons (L2&3: 81.9 ± 13.2 mV, L3c: 88.3 ± 7.5 mV, L5: 88.7 ± 18.2 mV, $p = 0.020$; Fig. 3d). In addition, the AP half-width was similar in L5 compared to L2&3 (L2&3: 1.8 ± 0.6 ms, L5: 1.8 ± 0.7 ms). However, the half-width of the AP was significantly longer in L3c pyramidal cells compared to L2&3 ($p = 0.029$, L3c: 2.2 ± 0.6 ms) and L5 pyramidal cells ($p = 0.022$, Fig. 3f).

The frequency–current relationships ($f$–$I$ curve) showed greater $f$–$I$ slopes for L5 relative to L2&3 and L3c neurons (Fig. 3g; at 300 pA, L2&3: 16.3 ± 12.2 Hz, L3c: 11.0 ± 12.2 Hz, L5: 21.8 ± 13.1 Hz, $p = 0.003$ between L2&3 and L5 neurons and $p = 0.001$ between L5 and L3c neurons). The current needed to elicit an AP was significantly lower in L5 neurons compared to L2&3 (L2&3: 162.1 ± 81.9 pA, L3c: 153.5 ± 66.6 pA, L5: 121.1 ± 85.2 pA, $p = 0.002$ between L2&3 and L5; Fig. 3h). We also note that these distributions were especially broad, particularly for L5 pyramidal cells, mirroring the large range in input resistances and diverse morphologies of these neurons (Fig. 1). L5 neurons showed significantly less spike frequency adaptation relative to L2&3 and L3c neurons (L2&3: 0.16 ± 0.16, L3c: 0.25 ± 0.16, L5: 0.11 ± 0.13, $p = 0.016$ between L5 and L2&3, $p = 0.001$ between L5 and L3c; Fig. 3i). Lastly, we identified a small number of bursting neurons (defined as those with instantaneous frequencies at rheobase >75 Hz) in our dataset (e.g., the L5 cell illustrated in Fig. 3c and further examples in Supplementary Fig. 6). Specifically, we found 14% of our recorded L2&3 pyramidal cells and 9.5% of L5 pyramidal cells showing bursting activity. The overall low number of bursting neurons in both superficial and deeper cortical layers of human neocortex are consistent with previous report of infrequent bursting in human neocortex[3].

### Subthreshold and suprathreshold frequency preference in human pyramidal cells across cortical layers.

Resonance is a common approach to characterize the frequency preferences of neurons and represents the net result of the interaction between passive and active properties[30]. Indeed, human pyramidal cells, and in particular those in deeper part of L3 as well as thick-tufted neurons in L5, can exhibit low-frequency subthreshold resonance[3,4]. These findings in human neurons are consistent

with studies in rodent cortex that describe the correlation between a large sag voltage and low-frequency resonance[31].

We examined subthreshold resonance in our recorded L2&3, L3c, and L5 pyramidal cells using a 20-s long frequency-modulated (or ZAP) current stimulus delivered at the RMP[20]. Our analysis revealed clear examples of resonant pyramidal cells in each of the three major layers we profiled (Fig. 4a, b) and identified a number of neurons that displayed a non-zero peak in their resonant frequency (fR; 27% of L2&3 neurons, 47% of L3c neurons, and 40% of L5 neurons) but considerably few neurons with fRs >2 Hz (Fig. 4c). We found there was a slight trend of a decrease in the 3 dB cutoff frequency in L5 relative to L2&3 neurons ($p = 0.050$, Fig. 4d). These results are generally consistent with recent evidence for greater subthreshold resonance in the deeper part of the supragranular layers of the human neocortex relative to more superficial neurons[4]. While we observed a smaller fraction of resonant cells than previous work, we note that our results correspond with the conclusion that human L2&3 pyramidal cells are most likely to have normalized impedance peaks at <2 Hz, while neurons with peaks at >4 Hz are quite rare. Possible explanations for the lower fraction of resonant cells in our data include our use of different experimental solutions than Kalmbach et al., as well as the possibility of inadvertent dendrite truncation in these experiments (see "Discussion"). Additionally, our neurons displayed a slightly more depolarized RMP[4], which is a determinant of observing resonance[32].

We further compared the frequency response characteristics of L2&3 and L5 pyramidal cells in response to suprathreshold ZAP current injections with example traces shown in Fig. 4e. We found that L5 neurons spike with greater fidelity to higher frequency stimuli (12–18 Hz) relative to L2&3. We did not observe a difference in frequency tracking at other frequency ranges (Fig. 4f), including frequencies at delta or theta (1–8 Hz). We note that the lack of direct correspondence between our subthreshold and firing rate resonance properties is not surprising, especially in light of theoretical and computational explorations that reveal the lack of a direct link between subthreshold and suprathreshold stimuli responses[33–36].

### Assessment of frequency-dependent gain reveals greater preference for delta and theta frequencies among L5 relative to L2&3 pyramidal cells.

To further investigate the suprathreshold frequency preference of human neurons, a key determinant of their participation in the amplification and/or generation of oscillations[13], we characterized the frequency-dependent gain [$G(f)$] and the mean phase shift of the spike response (Fig. 5a)[37] in a subset of neurons ($n = 8$ neurons each for L5 and L2&3) from an additional set of patients ($n = 5$). $G(f)$ was quantified via stimulating neurons with multiple trials of a frozen filtered white noise current stimulus. This measure captures distinct neuronal features compared to subthreshold or suprathreshold resonance: while resonance identifies the likelihood of a spike occurring from a drive at a particular frequency that is itself is suprathreshold, the frequency-dependent gain quantifies the phase preference of neuronal spiking as a function of frequency[38] from a noisy input that is relatively small[37]. Neurons with a high gain at a specific frequency are more likely to have a phase preference at that frequency than at other frequencies.

We found that both L2&3 and L5 neurons displayed peaks in $G(f)$ within the delta and theta frequency ranges (Fig. 5b). Both peaks were significantly more pronounced in L5 pyramidal cells compared to L2&3 ($p < 0.05$). Additionally, above 10 Hz, we found that L5 pyramidal cells displayed greater frequency-dependent gain than L2&3 neurons, corresponding with the

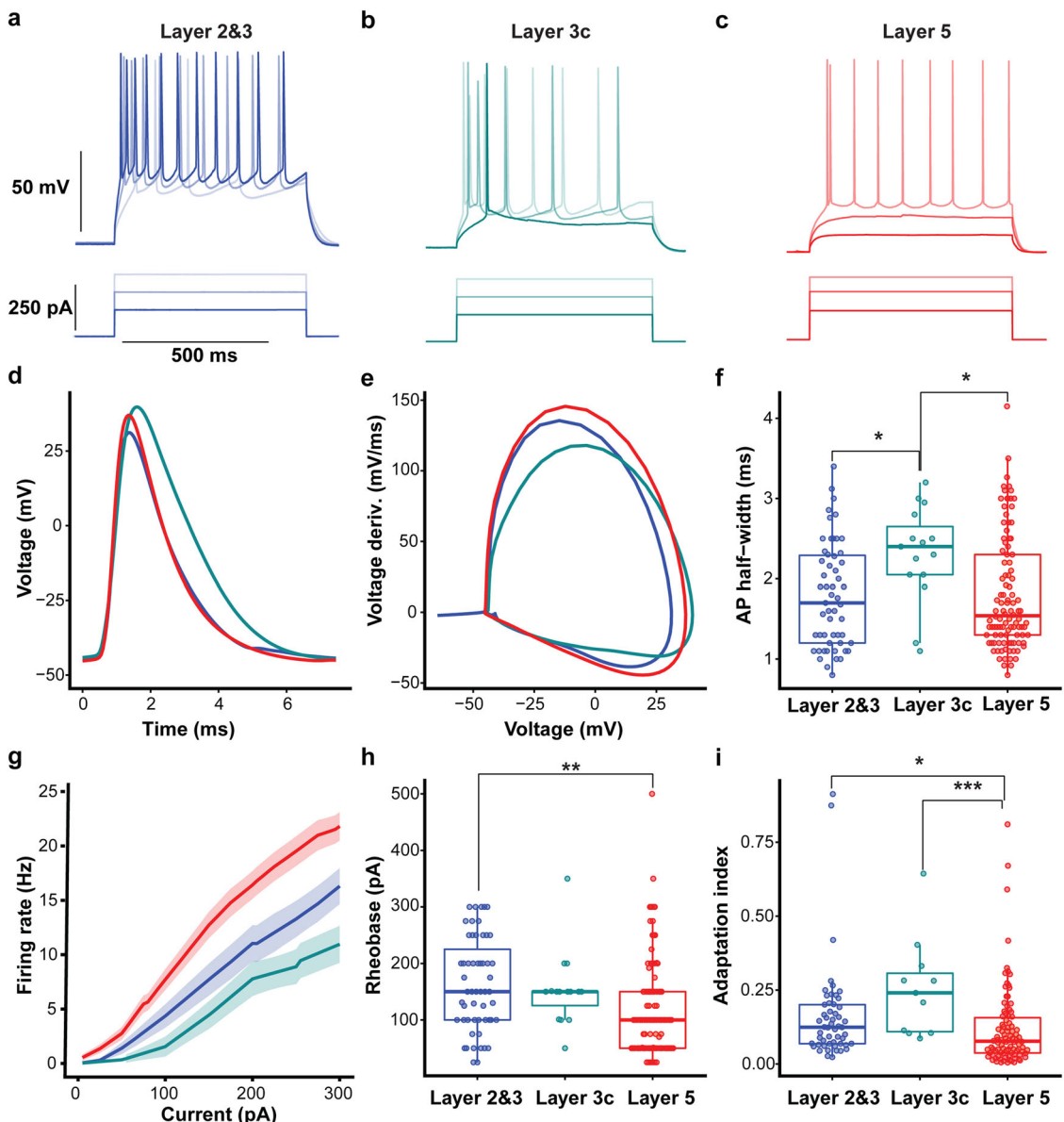

**Fig. 3 L5 pyramidal cells display higher frequency firing rate and less adaptation than L2&3 and L3c neurons. a–c** Example of L2&3, L3c, and L5 pyramidal cell voltage responses following depolarizing current injections (150, 250, and 300 pA, 600 ms). **d, e** Action potential waveform (**d**) and action potential phase plot (**e**) averaged over recorded pyramidal cells in each layer. **f** L5 pyramidal cells had a smaller action potential half-width compared to L2&3 and L3c neurons, which was not significant in comparison with L2&3 neurons. Half-width of action potential was higher significantly in L3c pyramidal cells compared with L2&3 and L5 neurons ($p = 0.029$ between L2&3 and L3c, $p = 0.022$ between L3c and L5). **g** Current versus firing rate relationships (FI curves), averaged over pyramidal cells recorded in each layer. Shaded bands indicate SEM. **h** L5 pyramidal cells needed less depolarizing current to display first action potential compared to L2&3 and L3c neurons ($p = 0.002$). **i** L5 pyramidal cells show less spike frequency adaption, quantified using the adaptation index measure, in comparison with L2&3 and L3c. ($p = 0.016$ between L2&3 and L5; $p = 0.001$ between L3c and L5). One-way ANOVA post hoc with Dunn's multiple comparison test were used for statistical comparison L2&3 ($n = 55$), L3c ($n = 15$), and L5 ($n = 104$). Data presented as mean ± SD in panels **f, h, i**. Boxplots in **f, h, i** denote interquartile range and whiskers denote data range excluding outliers. * indicates $p < 0.05$, ** indicates $p < 0.01$ and *** indicates $p < 0.001$. Source data are provided as a Source data file.

intuition from our suprathreshold ZAP results (Fig. 4f). The greater excitability and fidelity of L5 neurons was also evident in their phase curves and spike-triggered averages (STAs; Fig. 5c, d). L2&3 neurons demonstrated a greater lag in firing than L5 neurons, and their STAs were of larger amplitude with steeper slopes. This suggests that L2&3 neurons require larger inputs to trigger spikes, and when they do spike, they will lag behind L5 pyramidal cells if inputs are coincident.

To explore the contribution of $I_h$ to $G(f)$ for both L2&3 ($n = 3$, Fig. 5e) and L5 pyramidal cells ($n = 3$, Fig. 5f), ZD7288 (an

$I_h$ blocker) was applied to compare $G(f)$ before and after abolishing the $I_h$. We found that blocking $I_h$ predominantly abolished the delta peak in L5 neurons. These data indicate that human L5 pyramidal cells are better at tracking both delta and theta frequency inputs than superficial layer neurons (although L3c neurons were not tested in this way), and our (perhaps preliminary) explorations of the effect of ZD on the frequency-dependent gain provides evidence that the larger $I_h$ in L5 pyramidal cells plays an important role in their increased responsiveness to delta frequency inputs.

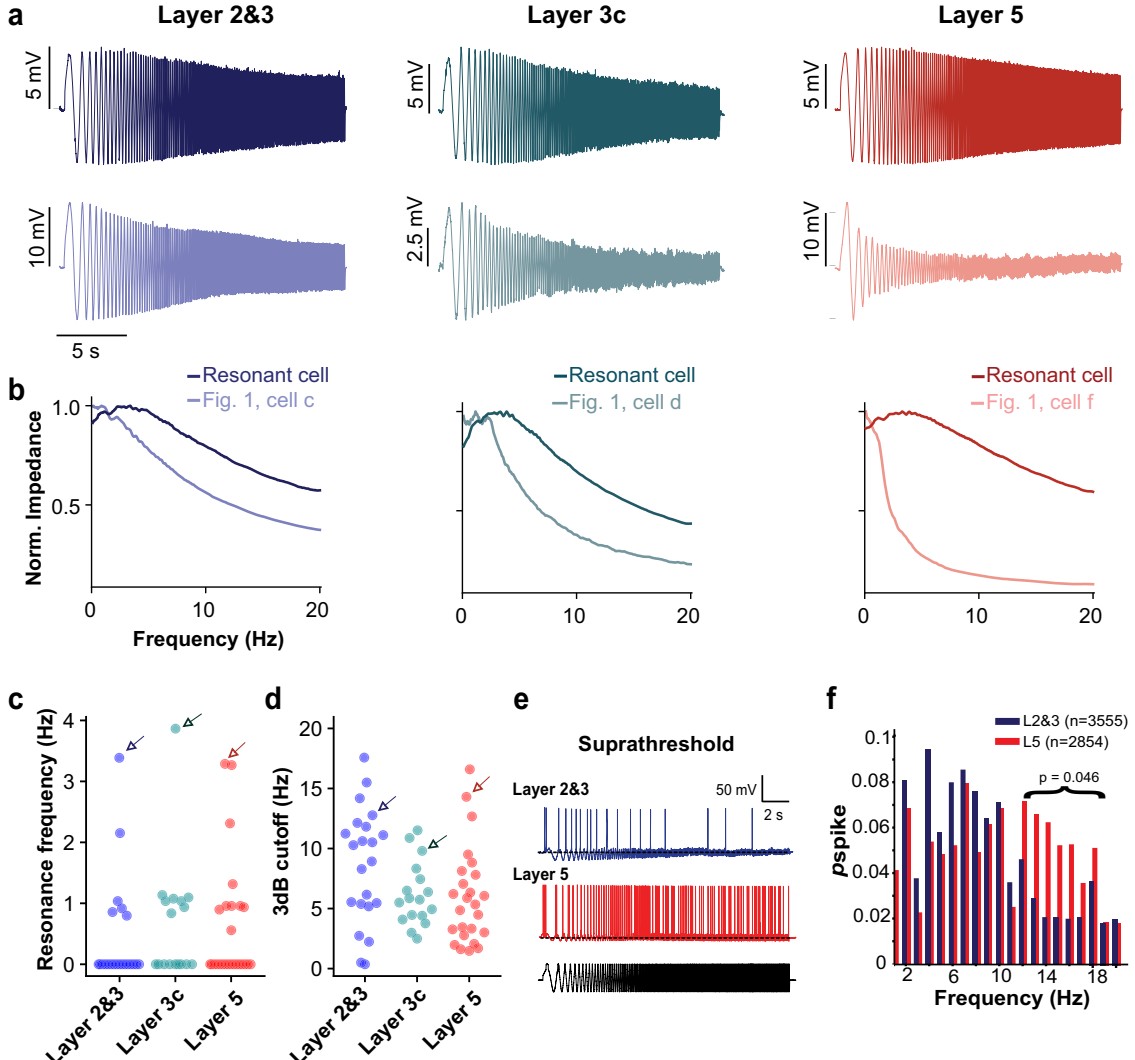

**Fig. 4 Subthreshold and suprathreshold resonance properties of L2&3, L3c, and L5 pyramidal cells. a** Example of voltage responses following injection of subthreshold frequency-modulated (ZAP) current delivered at resting membrane potential in L2&3, L3c, and L5 pyramidal cells. Voltage traces are averaged across five repeated trials. Top voltage trace indicates response of a cell displaying subthreshold resonance and bottom trace corresponds to example cell morphologies shown in Fig. 1. Cell labels are provided below in **b**. **b** Normalized impedance profiles for the voltage traces shown in **a**, with impedances normalized to a maximum value of one. **c**, **d** Resonance frequencies (**c**) and 3 dB cutoff frequencies (**d**) indicate that there are relatively few strongly resonant pyramidal cells in our dataset. Arrows in **c**, **d** correspond to resonant cells highlighted in **a**, **b**. $n = 23$, 17, and 25 cells in L2&3, L3c, and L5, respectively. **e** Example voltage responses following injection of suprathreshold ZAP current. **f** Spike response probability in response to suprathreshold current injection shows that L2&3 and L5 pyramidal cells track theta frequencies with greater reliability. The distributions were not significantly different (KS $p = 0.530$). Above 12 Hz, spike probabilities differed significantly between L2&3 and L5 ($p = 0.046$; two-sample Kolmogorov–Smirnov, L2&3 $n = 27$, L5 $n = 30$). Source data are provided as a Source data file.

**Considerable heterogeneity and overlap in electrophysiological features among pyramidal cells sampled across different layers of the human neocortex.** Given the large degree of variability among intrinsic electrophysiological features in the pyramidal cells in our study, we next sought to identify gradients or sub-clusters among these neurons. For example, in rodent neocortex, there is strong convergent evidence that pyramidal cells from L5 are split into two major subclasses, with neurons from L5a more likely to be regular spiking (RS), IT projecting. IT cells show slender tufted dendritic morphologies compared to pyramidal cells in L5b, which are more likely to be bursting, ET, and show thick-tufted dendritic morphologies[28,39]. In the human neo-cortex, there is evidence for this dichotomy based on tran-scriptomics data, although there are likely far fewer ET neurons than IT neurons in human L5 relative to the rodent

(Supplementary Fig. 9)[25], which to our knowledge has yet to be corroborated at the electrophysiological and morphological levels.

We addressed this question using dimensionality reduction techniques to arrange our neurons by similarity in multi-variate sets of electrophysiological features (L2&3: $n = 56$, L3c: $n = 14$, L5: $n = 103$ neurons). We specifically used uniform manifold approximation (UMAP)[40] using 14 subthreshold and suprathres-hold electrophysiological features that were consistently calcu-lated in the majority of pyramidal cells within our dataset (see "Methods"). We found some evidence for gradients and/or subclusters among the sampled neurons based on the input electrophysiological features (Fig. 6a). Upon further inspection, we found that a single major factor related to cell input resistance and rheobase current appeared to qualitatively define the major gradient underlying the differences in neurons highlighted by this

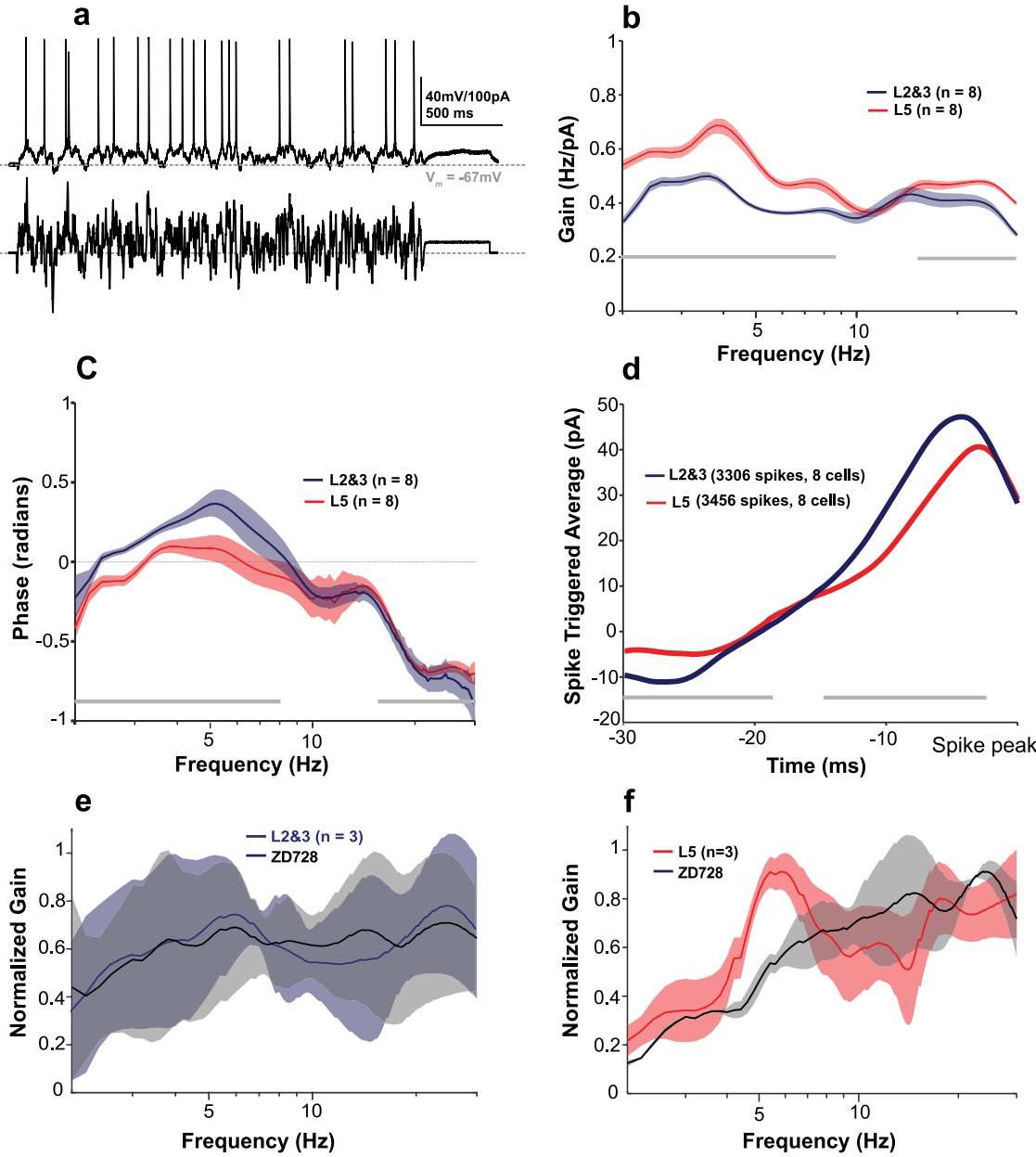

**Fig. 5 Human L5 neurons display greater gain at delta and theta frequencies than L2&3 pyramidal cells. a** Example of $V_m$ response of L5 pyramidal cell to 2.5 s of frozen filtered Gaussian white noise current injection. **b** Frequency-dependent gain $G(f)$ profile of L2&3 and L5 pyramidal cells over a wide range of frequencies. Both layers show two peaks around 2.5–10 and 12–16 Hz, which are more pronounced in L5 pyramidal cells compared to L2&3. Gray horizontal bars represent significant differences between groups ($p < 0.05$; Mann–Whitney $U$). **c** Phase shift of spiking relative to input stimulus. L2&3 pyramidal cells show positive phase in their mean phase shift profile, which represents a lag in L2&3 pyramidal cells compared to L5 pyramidal cells ($p < 0.05$; Mann–Whitney $U$). **d** Mean spike-triggered average (STAs) for L5 and L2&3 neurons. Difference in STAs indicate that L5 neurons require less current and instantaneous rate of current increase to initiate a spike (greater excitability). **e**, **f** Frequency-dependent gain profile (mean ± one standard deviation) of L2&3 ($n = 3$; RMP: −66.2 ± 2.9 mV, input resistance: 105 ± 26.46 MΩ) (**e**) and L5 ($n = 3$; RMP: −66.8 ± 3.1 mV, input resistance: 81.3 ± 12 MΩ) (**f**) pyramidal cells before and after $I_h$ blocker (ZD7288 10 μM). ZD7288 abolished the low-frequency peaks in L5 neurons with little change in frequency-dependent gain in L2&3 neurons. Source data are provided as a Source data file.

analysis (Fig. 6b). We were able to corroborate aspects of this unbiased analysis through inspection of pyramidal cell morphologies (where available) with neurons at one extreme of the gradient having the largest input resistances and the most simple morphologies (e.g., cell a and cell f from Fig. 1). Similarly, neurons with morphologies on the other side of the gradient tended to have lower input resistances and more complex morphologies such as the larger cell (cell g) shown in Fig. 1. This analysis also revealed that neurons throughout the gradient tend

to show bursting behavior (Fig. 6c) and that neurons with intermediate input resistances (such as those sampled from L3c) are more likely to display subthreshold resonance (Fig. 6d).

The other major finding of this analysis is that neurons from each of the layers we sampled were often inter-mixed in the low-dimensional space, with neurons from L2&3 often displaying very similar electrophysiological profiles to those sampled in L5. The pyramidal cells sampled from L3c were one exception, as these were present primarily at a single position in the low-dimensional

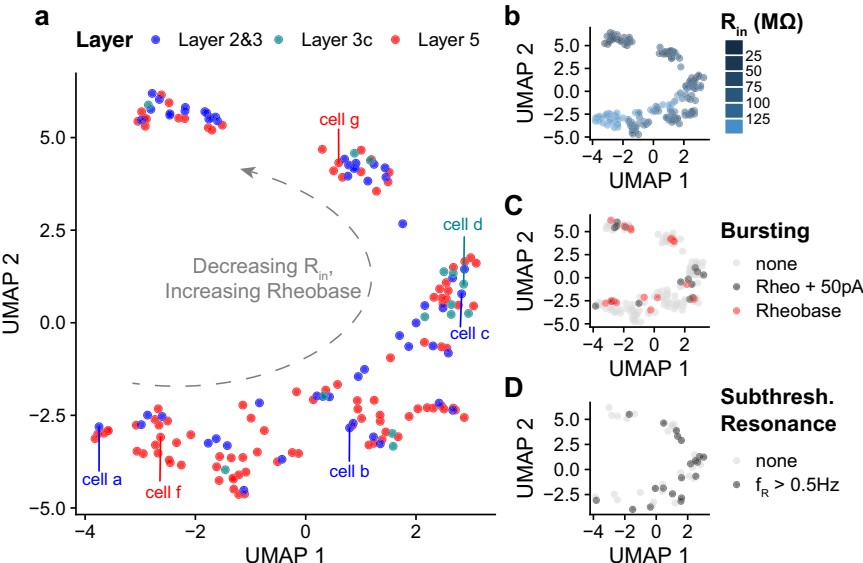

**Fig. 6 Dimensionality reduction reveals considerable intrinsic electrophysiological similarity and overlap between pyramidal cells recorded in different layers. a** Dimensionality reduction analysis performed on multivariate sets of electrophysiological features using uniform manifold approximation (UMAP). Neurons are arranged by similarity in intrinsic electrophysiological features (see "Methods" for list of features used in analysis). Cell counts: L2&3: $n = 56$, L3c: $n = 14$, L5: $n = 103$ neurons. Cell labels correspond to neurons whose morphologies are highlighted in Fig. 1. **b–d** Same as **a**, but neurons are colored by input resistance (**b**), bursting behavior (**c**), or subthreshold resonance (**d**). Bursting behavior in **c** is broken down by neurons that show no bursting, neurons that burst at rheobase current plus 50 pA, or neurons that burst at rheobase. Bursting is defined where the instantaneous firing rate is >75 Hz. Source data are provided as a Source data file.

space along with other neurons of intermediate input resistance. As we did not see strong evidence for a subtype of L5 neurons that were electrophysiologically distinct from those in L2&3 and other L5 neurons, we conclude that either we do not have L5 ET neurons present in our dataset (a strong possibility given their rarity, shown in Supplementary Fig. 9[25]) or that we are unable to distinguish them from intratelencephalic (IT) neurons using electrophysiological features alone (see "Discussion"). In summary, these findings are largely consistent with recent transcriptomics data[25] that strongly suggest that layer membership is not particularly informative dimension regarding pyramidal cell-type diversity in human cortex, as many discrete transcriptomically defined cell types do not obey strict laminar boundaries.

**Putative interneurons recorded in L5 show greater amounts of sag and subthreshold resonance relative to L2&3.** Our dataset also included several putative GABAergic interneurons (examples in Supplementary Fig. 7a; L2&3: $n = 10$, L5: $n = 14$). We were able to distinguish putative interneurons from pyramidal cells by their AP characteristics (Supplementary Fig. 7b, c), large maximal firing rates, and typically large spike after-hyperpolarization amplitudes (see Supplementary Fig. 8 for morphological corroboration for one putative interneuron from our dataset). We found that the set of putative interneurons in L5 had larger sag voltage amplitudes compared to putative interneurons in L2&3 (L2&3: $1.2 \pm 1.1$ mV, L5: $4.5 \pm 3.7$ mV, $p = 0.001$; Supplementary Fig. 7d). Moreover, L5 putative interneurons had significantly larger rebound depolarization amplitudes relative to L2&3 putative interneurons (L2&3: $1.7 \pm 2.2$ mV, L5: $5.2 \pm 3.2$ mV, $p = 0.003$; Supplementary Fig. 7e). Moreover, while there was a relatively small number of interneurons that were characterized with ZAP current injection (examples in Supplementary Fig. 7f, g), we noticed a comparatively large fraction of L5 putative interneurons that displayed a non-zero peak in their fR (3 of the 6 neurons) compared to L2&3 (1 of the 3 neurons) (Supplementary Fig. 7h). Although a small number of

neurons and thus requiring corroboration, these data are consistent with prior reports of strong subthreshold resonance activity in human cortical GABAergic interneurons[5] and in rodent hippocampal interneurons[41].

## Discussion
Guided by our previous work implicating deep layer human pyramidal cells in driving coherent low-frequency oscillations in human neocortex, we sought to characterize how the electrophysiological differences between deep and superficial human pyramidal cells might inform the distinct role deep layer cells might play in cortical oscillations. We summarize four major findings from this work.

First, considering each broad cortical layer as a group, we found that there is a gradient of increased excitability from superficial to deeper layer pyramidal cells, with L2&3 pyramidal cells demonstrating more hyperpolarized resting potentials, lower input resistances and larger rheobase required for spike generation, enhanced spike frequency adaptation, and steeper and larger amplitude STAs. Along most of these features, the neurons sampled from L3c were often at an intermediate point between L2&3 and L5.

Second, we found enhanced sag and $I_h$-related features in L5 neurons relative to L2&3 neurons, again with L3c neurons intermediate between these groups. $I_h$ appeared to be one of the major contributors to the prominent rebound depolarization and rebound spiking in human L5 neurons, as ZD7288 significantly reduced both. $I_h$ also contributed to enhanced frequency-dependent gain at delta in L5 relative to L2&3 pyramidal cells being abolished using the $I_h$ blocker ZD7288. Voltage-clamp data suggested that this prominence of $I_h$ in L5 pyramidal cells was due to increased channel number and not differences in kinetics of $I_h$ channels. Intriguingly, we found anecdotal evidence that $I_h$ appears more prominent in L5 relative to L2&3 putative GABAergic interneurons and that this might contribute to enhanced resonant activity in these neurons.

Third, while we identified numerous pyramidal cells displaying non-zero resonant peaks in each cortical layer, we found resonance at frequencies >2 Hz to be a largely uncommon feature. This corresponds with previous reports finding cells, albeit rarely, exhibiting resonance at > 4 Hz in the deeper parts of L3[4] and in larger, thick-tufted neurons in L5[3].

Fourth, we found a great degree of electrophysiological heterogeneity among pyramidal cells sampled within each cortical layer. Consistent with recent reports describing the variability in morpho-electric and transcriptomic subtypes of L2 and L3 pyramidal cells[4,26,27], we found a similar and potentially greater amount of electrophysiological variability among human L5 pyramidal cells. Such biophysical variability among neurons of the same cell type is an increasingly recognized and computationally important aspect of neural circuits[42–44]. Our sampling of L5 pyramidal neurons is consistent with recent transcriptomic evidence suggesting that the vast majority of excitatory neurons in human L5 middle temporal gyrus (MTG) are IT projecting[25]. Another source of variability arises from the known dichotomy between L5 ET and IT projecting neurons, which have been extensively characterized in rodents[28,39]. This source of variance is less likely a contributor to the biophysical variability we observe since IT cells make up only 0.6% of glutamatergic neurons in human L5 (Supplementary Fig. 9). Thus, unlike a recent report[3] that targeted L5 thick-tufted neurons, the majority of the L5 neurons sampled here with their relatively high input resistances (>90 MΩ) suggests that we recorded primarily from the abundant thin-tufted IT pyramidal cells. We note that disentangling these hypotheses requires further corroboration and will likely require the use of emerging tools such as Patch-seq[26] to merge cell taxonomies along multiple modalities.

Given our findings of greater $I_h$ in L5 neurons, we were initially surprised that a larger number of neurons did not demonstrate a peak in subthreshold resonance at frequencies >4 Hz. For example, a previous report by Kalmbach et al. suggested that $I_h$ contributes to prominent subthreshold resonance in deep L3 human pyramidal cells[4]. Similarly, somatic subthreshold resonance has also been reported in human L5 thick-tufted neurons[3]. In addition, previous work in rodents has shown that L5 pyramidal cells are endowed with subthreshold resonance[45–48]. However, our findings are not inconsistent with previous results in the human setting. In particular, the work of Kalmbach et al. reports notably few cells in L2&3 exhibiting subthreshold resonance at > 4 Hz, as well as many cells exhibiting no resonant peak at all when held at a common membrane potential of −65 mV[4]. Similarly, we note that, while Beaulieu-Laroche et al. reported strong subthreshold resonance among L5 pyramidal cells, these recordings were intentionally targeted toward the largest cells in L5 (and are thus likely to reflect characterization of the rare ET cells[3,25]).

It is worth emphasizing that subthreshold resonance is a complex dynamic not dictated solely by the amount of $I_h$ present in a cell, which likely explains the minor differences in proportions of resonant cells reported in our work and that of Kalmbach et al.[4]. In fact, interactions between $I_h$[46], persistent Na+ current ($I_{NaP}$), $K_{IR}$ (instantaneously activating, inwardly rectifying K+ current)[47], M-current[20], and passive properties[48] are all thought to influence this dynamic. Hippocampal oriens-lacunosum molecular interneurons[41], CA1 pyramidal cells, oriens-radiatum interneurons[41], and inferior olivary neurons[49] are all examples of cells where subthreshold resonance is not solely and/or directly driven by $I_h$. When viewed in concert with our recent detailed computational investigation of the relationship between $I_h$ and subthreshold resonance[32], our current results serve to highlight that a prominent $I_h$ is not always sufficient to drive subthreshold resonance. With subthreshold resonance not observed as a general feature of L5 pyramidal cells, we sought other biophysical features that might explain why L5 cells appear to drive interlaminar theta coherence. Recently, a putative DCM[13] has been proposed to underlie the interlaminar nested delta–theta oscillations observed in rodents. This DCM posits intrinsically bursting (IB) neurons in L5 neurons as central actors in generating deep layer activity that drives superficial theta oscillations[50]. Although the electrophysiological signature and experimental conditions studied in Carracedo et al. were different to ours in human cortical slices[9], it is instructive to relate our findings to what was observed in rat neocortex. Carracedo et al. demonstrated that delta oscillations likely occur due to tonic drive to the dendrites of IB neurons in superficial layers[50]. This tonic drive causes the IB neurons to discharge bursts at delta frequencies (~2 Hz). IB neurons are unique in that, in addition to their subcortical targets, they primarily synapse locally within deep layers on L5 RS IT neurons, unlike L5 RS neurons that project axons both locally and to L2&3[39,51,52]. The RS neurons are thus driven by periodic barrages at delta frequencies and discharge doublets with each IB burst, thus generating "theta" frequency output at double the L5 delta frequency, which is then transmitted to superficial layers. The sinks generated in the superficial layers thus occur at theta frequency, driving local excitability in L2&3 with the resultant increase in excitatory drive to L5 IB dendrites starting the cycle anew.

Our results demonstrating 4 and 8 Hz peaks in $G(f)$ for L5 RS neurons, interpreted in the context of the above findings by Carracedo et al., provides a plausible mechanism for the theta activity (~8 Hz) we observed in vitro[53] and that is ubiquitously observed in the human brain[17]. It is important to note that the theta generated by RS neurons described by Carracedo et al. arises from the doublet generated in response to each cycle of delta, and thus why theta (~4 Hz) was twice the frequency of the observed delta (~2 Hz) in their work. The double peak in $G(f)$ we observe in human L5 RS neurons implies that RS neurons are tuned to both 4 and 8 Hz activity, and not surprisingly the 8 Hz peak in $G(f)$ is similar to the frequency at which interlaminar coherence was observed in human slices[9] and twice the frequency of the low-frequency peak in $G(f)$. Our ZD data further supports this relationship between the delta and theta peaks, where the delta peak in $G(f)$ in a different subset of neurons was ~5.5 Hz and the "theta" peak was at ~11 Hz. That blocking $I_h$ abolished the delta peak suggests that $I_h$ tunes L5 RS neurons to track IB output, which in turn generates theta (double the frequency of delta) output. Interpreted together, our frequency-dependent gain and ZAP results suggest that $I_h$ may not be a direct "cause" of cortical oscillations at theta (~8 Hz) but rather tune RS cells to follow with great fidelity the IB output at delta (see Supplementary Fig. 10 for this DCM).

An obvious difference between our previous human slice work[9] and that of Carracedo et al. is that we observed robust deep layer theta, although theta was still more prominent in the superficial layers. One possible explanation is that it has been shown that, in human L2&3, a single AP generates long-lasting reverberant activity through rebound excitation that lasts an order of magnitude longer than in the rodent[7]. Thus, it is possible that such reverberant activity as well exists in L5 resulting in greater gain in local L5 cortical circuits that amplifies theta activity through both synaptic activation and the theta peak in $G(f)$. This conjecture is further supported by our observation that putative L5 interneurons demonstrate greater rebound depolarization than L2&3 neurons and thus are likely able to amplify network activity within L5 potentially beyond what was observed in L2&3[7]. Future experiments are needed to explore whether human cortical circuitry is arranged like that of the rodent, specifically as it relates to interlaminar and intralaminar connectivity.

It bears acknowledging that experimental limitations might have influenced our observation of subthreshold resonance. The increased density of the HCN channel in dendrites may result in resonance being observed better in the dendrites compared to the soma[3], and despite best practice controls the possibility remains that truncated dendrites might affect resonance. Truncation of layer 5 pyramidal cell dendrites is a common and unavoidable issue due to longer apical dendrite length in the human neocortex (2 mm)[1,3]. However, there is strong evidence for increased dendritic compartmentalization in large human neurons with distal inputs attenuating strongly toward soma[3], and one would expect this effect to mitigate any effects of dendritic truncation on subthreshold resonance.

An important caveat when interpreting these findings is that these data are exclusively collected from neurosurgical patients undergoing surgery for drug-resistant epilepsy or for resection of brain tumors. We have been careful to only record from unaffected (non-epileptogenic) neocortical tissue. Nevertheless, it is unclear how these diseases (or their pharmacological treatment regimes) might contribute to compensatory changes at the level of cortical neuron physiology. Notwithstanding that epilepsy patients represent the primary source of viable human tissue for in vitro human studies[1,3,4,25], our data are comparable to these human cell-typing efforts, since our inclusion criteria for our samples is consistent with these human studies. Additionally, by comparing our findings to analogous human neuronal datasets collected by other groups, we are confident our results are comparable to similar human cell-type characterizations. Lastly, the most ubiquitous source of tissue from this study (MTG of epilepsy patients) demonstrates seeming transcriptomic "normalcy" when compared to post-mortem specimens from the MTG[25].

This report reflects one of the largest studies of the electrophysiological diversity of human neocortical pyramidal cells to date, contributing to our growing understanding of human L5 pyramidal cells[3] and serving to put the unique characteristics of these neurons into context with the better understood superficial layer pyramidal cells. Specifically, our unbiased sampling strategy of L5 cells complements the targeted characterization of large, thick-tufted L5 human pyramidal cells recently reported[3]. Given the rare opportunity to perform experiments in live human tissue, our work also represents an extremely valuable opportunity to compare findings with the limited existing literature on electrophysiological properties of human cortical neurons.

Moving forward, it will be essential to reconcile these electrophysiological and morphological data with the emerging consensus of neocortical cell-type diversity based on single-cell transcriptomics[3,4,25,54] and how these features contribute to the unique emergent properties of human cortical circuits. Furthermore, little is known about the connectivity within human cortical circuits: is human interlaminar and intralaminar connectivity similar to rodents, and how do cellular properties contribute to the signatures observed in mesoscopic and macroscopic recordings? Answering these questions will require multiscale inquiries of human cortical micro-circuits and in silico experiments to understand the divergent properties of human circuits, with the tools for such inquiries only now becoming available.

## Methods

**Human brain slice preparation**. Written informed consent was obtained from all study participants to use their tissue as well as to share the acquired electrophysiological data and anonymized demographic information—including age, sex, years of seizure, diagnosis, and antiepileptic drug treatment—as stated in the research protocol. In accordance with the Declaration of Helsinki, approval for this study was received by the University Health Network Research Ethics board. Sixty-one patients, age ranging between 19 and 63 years (mean age: 37.1 ± 1.8 years),

underwent a standard anterior temporal lobectomy[55] or tumor resection from the frontal or temporal lobe[56,57] under general anesthesia using volatile anesthetics.

The surgery involved resecting the first 4.5 cm of neocortex using sharp dissection and local cooling with ~4 °C TissueSol®. Immediately following surgical resection, the cortical block was submerged in an ice-cold (~4 °C) cutting solution that was continuously bubbled with 95% $O_2$–5% $CO_2$ containing (in mM) sucrose 248, KCl 2, $MgSO_4.7H_2O$ 3, $CaCl_2.2H_2O$ 1, $NaHCO_3$ 26, $NaH_2PO_4.H_2O$ 1.25, and D-glucose 10. The osmolarity was adjusted to 300–305 mOsm. The total duration, including slicing and transportation, was kept to a maximum of 20 min[57]. Transverse brain slices (400 μm) were obtained using a vibratome (Leica 1200 V) in cutting solution. Tissue slicing was performed perpendicular to the pial surface to ensure that pyramidal cell dendrites were minimally truncated[4,56]. The cutting solution was the same as used for transport of tissue from operation room to the laboratory. After sectioning, the slices were incubated for 30 min at 34 °C in standard artificial cerebrospinal fluid (aCSF) (in mM): NaCl 123, KCl 4, $CaCl_2.2H_2O$ 1, $MgSO_4.7H_2O$ 1, $NaHCO_3$ 26, $NaH_2PO_4.H_2O$ 1.2, and D-glucose 10, pH 7.40. All aCSF and cutting solutions were continuously bubbled with carbogen gas (95% $O_2$–5% $CO_2$) and had an osmolarity of 300–305 mOsm. Following this incubation, the slices were maintained in standard aCSF at 22–23 °C for at least 1 h, until they were individually transferred to a submerged recording chamber.

For a subset of experiments designed to assess frequency-dependent gain, slices were prepared using the NMDG protective recovery method[58]. The slicing and transport solution was composed of (in mM): NMDG 92, KCl 2.5, $NaH_2PO_4$ 1.25, $NaHCO_3$ 30, HEPES 20, Glucose 25, Thiourea 2, Na L-ascorbate 5, Na-Pyruvate 3, $CaCl_2.4H_2O$ 0.5, and $MgSO_4.7H_2O$ 10. The pH of NMDG solution was adjusted to 7.3–7.4 using hydrochloric acid and the osmolarity was 300–305 mOsm. Before transport and slicing, the NMDG solution was carbogenated for 15 min and chilled to 2–4 °C. After slices were cut (as described above), they were transferred to a recovery chamber filled with 32–34 °C NMDG solution and continuously bubbled with 95% $O_2$–5% $CO_2$. After 12 min, the slices were transferred to an incubation solution containing (in mM): NaCl 92, KCl 2.5, $NaH_2PO_4.H_2O$ 1.25, $NaHCO_3$ 30, HEPES 20, Glucose 25, Thiourea 2, Na L-ascorbate 5, Na-Pyruvate 3, $CaCl_2.4H_2O$ 2, and $MgSO_4.7H_2O$ 2. The solution was continuously bubbled with 95% $O_2$–5% $CO_2$. After 1-h incubation at room temperature, slices were transferred to a recording chamber and continuously perfused with aCSF containing (in mM): NaCl 126, KCl 2.5, $NaH_2PO_4.H_2O$ 1.25, $NaHCO_3$ 26, Glucose 12.6, $CaCl_2.2H_2O$ 2, and $MgSO_4.7H_2O$ 1. We emphasize that these experiments were performed with excitatory (APV 50 μM, Sigma; CNQX 25 μM, Sigma) and inhibitory (Bicuculline 10 μM, Sigma; CGP-35348 10 μM, Sigma) synaptic activity blocked. These blockers are only used in these experiments, highlighted in Fig. 5.

**Electrophysiology recordings and intrinsic physiology feature analysis**. For recordings, slices were transferred to a recording chamber mounted on a fixed-stage upright microscope (Olympus BX51WI upright microscope; Olympus Optical Co., NY, USA and Axioskop 2 FS MOT; Carl Zeiss, Germany). Slices were continually perfused at 4 ml/min with standard aCSF at 32–34 °C. Cortical neurons were visualized using an IR-CCD camera (IR-1000, MTI, USA) with a ×40 water immersion objective lens. Using the IR-DIC microscope, the boundary between layer 1 and 2 was easily distinguishable in terms of cell density. Below L2, the sparser area of neurons (L3) were followed by a tight band of densely packed layer 4 (L4) neurons. L4 was followed by a decrease in cell density (L5). In general, we did not annotate different neurons recorded from L2 versus those recorded from L3, except when explicitly mentioned. In this study, we use the terminology "L2&3" to highlight that these layers are distinct in the human cortex, rather than indistinguishable as in the rodent cortex. Cells specifically targeted in deep L3 are further distinguished by being denoted as coming from "L3c".

Patch pipettes (3–6 MΩ resistance) were pulled from standard borosilicate glass pipettes (thin-wall borosilicate tubes with filaments, World Precision Instruments, Sarasota, FL, USA) using a vertical puller (PC-10, Narishige). Pipettes were filled with intracellular solution containing (in mM): K-gluconate 135, NaCl 10, HEPES 10, $MgCl_2$ 1, $Na_2ATP$ 2, and GTP 0.3, pH adjusted with KOH to 7.4 (290–309 mOsm). In a subset of experiments, the pipette solution also contained biocytin (3–5%). Whole-cell patch-clamp recordings were obtained using a Multiclamp 700 A amplifier, Axopatch 200B amplifier, and pClamp 9.2 and pClamp 10.6 data acquisition software (Axon instruments, Molecular Devices, USA). Subsequently, electrical signals were digitized at 20 kHz using a 1320X digitizer. The access resistance was monitored throughout the recording (typically between 8 and 25 MΩ), and neurons were discarded if the access resistance was >25 MΩ. The liquid junction potential was calculated to be −10.8 mV and was not corrected.

Data were analyzed offline using Clampfit 10.7, Python, MATLAB, and R software. The RMP was measured after breaking into the cell (IC = 0). The majority of the intrinsic electrophysiological features reported here were calculated using Python IPFX toolbox (https://github.com/AllenInstitute/ipfx/) with default parameter settings[59]. The input resistance and membrane time constant were calculated using hyperpolarizing sweeps between −50 and −200 pA. Single AP features, like the AP threshold, peak, width at half-max, and the upstroke–downstroke ratio, were calculated using the first spike at rheobase. The adaptation index, average firing rate, and inter-spike intervals (first, mean, median, coefficient of variation) were defined using the "hero" sweep with default parameters (defined as the sweep between 39 and

61 pA greater than the rheobase). Sag amplitude and sag ratio were defined in response to hyperpolarizing current pulses (600 ms duration, 0 to −400 pA, 50 pA steps). The sag ratio was calculated as the difference between the minimum value and the steady state divided by peak deflection during hyperpolarization current injection. The rebound depolarization amplitude was calculated as the difference between the steady-state voltage and the maximum depolarization potential. We also observed the presence or absence of rebound spiking following the injection of hyperpolarization current steps (−400 pA). The $I_h$ blocker ZD7288 (10 μM, Sigma Aldrich) was applied to confirm pharmacological evidence for $I_h$.

Bursting neurons were defined as those where the instantaneous frequency (determined by the first inter-spike interval at rheobase) was >75 Hz. We identified putative interneurons within our dataset by manually assessing each cell's maximum firing rates, spike widths, and after-hyperpolarization amplitudes. Putative interneurons that we identified using these criteria typically had spike half-widths <1 ms, after-hyperpolarization amplitudes >10 mV, and maximum firing rates >75 Hz[60]. We note that one limitation of this intrinsic feature-based identification criteria is the relative inability to identify vasoactive intestinal peptide interneurons (VIP) and other caudal ganglionic eminence-derived interneurons using intrinsic electrophysiological criteria alone[61].

**Voltage-clamp characterization of $I_h$ and $I_{tail}$.** To characterize $I_h$, 600-ms-long voltage-clamp steps were used in −10 mV increments, down to −140 mV from a holding potential of −60 mV. In order to measure $I_h$ amplitude, the difference between the steady state at the end of the holding potential and the maximum current was determined. The $I_{tail}$ was quantified as the difference between peak amplitude of residual current at the end of each holding potential and the steady-state current from holding potentials of −140 to −60 mV. A single- or double-exponential model, fitted to the various currents recorded, was used to calculate the time constants of $I_h$ in order to determine the kinetics of $I_h$. To measure the voltage sensitivity of $I_h$ in L2&3 and L5 pyramidal cells, the membrane potential evoking half-maximal activation of $I_h$ (V$_{50}$) was obtained by fitting the $I_h$ activation to a Boltzmann sigmoid function using GraphPad 6 (GraphPad, San Diego, CA, USA). In experiments to quantify $I_h$, the sodium channel blocker tetrodotoxin (1 μM; Alomone Labs) to block voltage-gated sodium currents, $CoCl_2$ (2 mM; Sigma-Aldrich) to block voltage-sensitive calcium currents, and $BaCl_2$ (1 mM; Sigma-Aldrich) to block inwardly rectifying potassium current were added to the bath solution. We note that space clamp issues limit the precise quantification of the $I_h$[29].

**Subthreshold resonance and spike probability analyses.** To assess subthreshold and suprathreshold resonance properties, a frequency-modulated sine wave current input (ZAP/chirp) was generated ranging from 1 to 20 Hz, lasting 20 s[31] with a sampling rate of 10 kHz. This current waveform was then injected using the custom waveform feature of Clampex 9.2 and Clampex 10.2 (Axon Instruments, Molecular Devices, USA). The subthreshold current amplitude was adjusted to the maximal current that did not elicit spiking.

For determining subthreshold resonance, only trials without spiking were utilized for analysis. Analyses were performed using in-house Python scripts adapted from Kalmbach et al.[4]. The impedance profile of the cell was computed by taking the ratio of the voltage over current in the frequency domain obtained with the fast Fourier transform. Window averaging was then applied to smooth the impedance profile of the cell. The impedance profiles were then averaged over several trials (up to five) to obtain the mean impedance profile of the cell. The frequency point with the highest impedance is the center frequency while the frequency point with half of the center impedance is the 3 dB cut-off frequency. Resonant neurons were defined as those with fRs >0.5 Hz, the lowest frequency tested here.

To analyze responses to suprathreshold frequency-modulated sinusoidal current, spiking probability as a function of input frequency was assessed using suprathreshold current stimulation. The suprathreshold current was set by gradually increasing the amplitude of the ZAP function input by adjusting the gain of the stimulus until the first spike was elicited. Ten traces per cell were utilized to obtain the probability of spiking as a function of frequency. Since the instantaneous frequency is known from the current input, each AP could be assigned a frequency at which it occurred. To create the spike probability density function for each cell type, the frequencies at which individual spikes occurred were pooled, and a histogram was generated and divided by the total number of spikes. To compare spike probability density functions between cell types, the distributions were compared using a two-sample Kolmogorov–Smirnov test (kstest2.m).

**Multi-variate electrophysiological feature analysis.** We used a dimensionality reduction approach to visualize similarities in recorded neurons according to multi-variate correlations in measured electrophysiology features. We specifically used the UMAP function and library implemented in R with default parameter settings[40]. We defined each recorded cell using feature vectors constructed from a set of 14 electrophysiological features that were reliably calculated in most characterized neurons. We specifically used the following subthreshold features: RMP, input resistance, membrane time constant, sag ratio, and sag amplitude. We

additionally used the following suprathreshold features of the first AP at rheobase: AP threshold, amplitude, half-width, upstroke–downstroke ratio, after-hyperpolarization amplitude, rheobase, and latency to first spike. We also used the following spike train features: slope of f–I curve and average spiking rate at the hero sweep stimulus.

**Frequency-dependent gain.** Following a similar methodology of Higgs et al.[37], frequency-dependent gain was computed using 30 trials (inter-trial interval = 20 s) of a 2.5-s duration current injection stimulus of frozen white noise convolved with a 3-ms square function[62]. This measure identifies the likelihood of the neuron spiking in phase with an oscillatory input that is small relative to the overall input to the cell, distinct from analysis of the neuron's activity in response to a suprathreshold ZAP input[37]. The amplitude (a.k.a. variance) of the current injection stimulus was scaled to elicit spike rates of >5 Hz, the typical firing rate for cortical pyramidal cells[63]. In addition to increasing the noise variance, a steady amount of direct current was required[37] to elicit spiking, which was delivered as various amplitude steps were added to the noisy current input. Peaks detected in the voltage time series with overshoot >0 mV were taken to be the occurrence of an AP. The time varying firing rate $r(t)$ was given by:

$$r(t) = \begin{cases} \frac{1}{\Delta t} & \text{Where spike detected} \\ 0 & \text{Where no spike detected} \end{cases} \quad (1)$$

The stimulus–response correlation ($c_{sr}$) and the stimulus autocorrelation ($c_{ss}$) were calculated in the following fashion:

$$c_{sr}(\tau) = \langle s(t)\, r(t+\tau)\rangle \quad (2)$$

$$c_{ss}(\tau) = \langle s(t)\, s(t+\tau)\rangle \quad (3)$$

where $\tau$ is the time difference and the stimulus $s(t)$ is $I_{noise}(t)$. After windowing the $c_{sr}(\tau)$ and $c_{ss}(\tau)$ functions (see below), the complex Fourier components $C_{sr}(f)$ and $C_{ss}(f)$ were obtained, and the frequency-dependent gain and the average phase shift were calculated with $\sigma = 1/f$, in order to ensure that the spectral estimates were not dominated by noise. The gain ($G(f)$) and the phase ($\varphi(f)$) are:

$$G(f) = \frac{|C_{sr}(f)|}{|C_{ss}(f)|} \quad (4)$$

$$\varphi(f) = \text{atan}\frac{[\text{Im}[C_{sr}(f)]]}{[\text{Re}[C_{sr}(f)]]} \quad (5)$$

where Re and Im refer to the real and imaginary parts of each Fourier component. $\varphi$(f) was then corrected using the peak time ($\tau_{delay}$) of $c_{sr}(\tau)$[37].

For statistical testing, individual gains or $G(f)$s for each cell (30 trials/cell) from neurons with spike rates above 5 Hz were pooled for each cell type. To compare between cell types, Mann–Whitney U (ranksum.m) was used to obtain a p value at each frequency (2–100 Hz in 0.2 Hz steps). The p values were the false discovery rate corrected with an alpha = 0.01[64].

**Histological methods.** During electrophysiological recording, biocytin (3–5 mg/ml) was allowed to diffuse into the patched neuron; after 20-45 min, the electrodes were slowly retracted under visual guidance to maintain the quality of the seal and staining. The slices were left for another 10–15 min in the recording chamber to washout excess biocytin from extracellular space, then transferred to 4% paraformaldehyde and kept at 4 °C for 24 h.

Subsequently, the slices were washed and transferred into phosphate-buffered saline (PBS) solution (0.1 mM). To reveal biocytin, slices were incubated in blocking serum (0.5% bovine serum albumin, 0.5% milk powder) and 0.1% Triton X-100 in PBS for 1 h at room temperature. Finally, slices were incubated with streptavidin-conjugated Alexa Fluor 594 (1:300) overnight at 4 °C. Then slices were rinsed with PBS and mounted on the slide using moviol (Sigma-Aldrich). Imaging was done using a Zeiss LSM710 Multiphoton microscope. Reconstructions were performed using the IMARIS software (Bitplane, Oxford Instrument Company).

**Statistical analyses.** Statistical analyses and plotting were performed using GraphPad Prism 6. Data are presented in the text as mean ± SD unless otherwise noted. Unless stated otherwise, a standard threshold of $p < 0.05$ was used to report statistically significant differences. One-way analysis of variance post hoc with Dunn's multiple comparison test were used for statistical comparison. The non-parametric Mann–Whitney test was used to determine statistical differences between the two groups. Wilcoxon matched-pairs signed rank test was used for paired comparison between the two groups.

**Reporting summary.** Further information on research design is available in the Nature Research Reporting Summary linked to this article.

## Data availability
Source data are provided with this paper.

## Code availability

Computational code for custom analyses are available at the following GitHub repositories: https://github.com/stripathy/valiante_ih (R) and https://github.com/stripathy/valiante_lab_abf_process (Python).

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

## Acknowledgements

We are immensely grateful to our neurosurgical patients and their families for consenting to the use of their tissue samples for research. We thank Dr. Gelareh Zadeh and Dr. Mark Bernstein for their assistance in obtaining brain tissue samples and Victoria Barkley and Marjan Rafiee for assistance in compiling demographic and chart information. We thank Sara Mahallati and Iliya Weisspapir for assistance in tissue preparation. We thank Brian Kalmbach for helpful conversations that informed our analysis of subthreshold resonance targeting of Layer 3c pyramidal cells. We thank Frances K Skinner, Etay Hay, Wesley Sacher, Dene Ringuette, Xiao Luo, and Jasmine Bell for their critical comments on the manuscript. We thank James Jonkman (Advanced Optical Microscopy Facility, University Health Network) for acquiring the two-photon images. We acknowledge generous support from the Centre for Addiction and Mental Health Discovery Fund, Kremblin Foundation, Krembil Brain Institute Fund, National Institute of Health, and Kavli Foundations.

## Author contributions

Conception or design of the experiments: H.M.C., T.A.V.; data collection: H.M.C., L.W.; data analysis and interpretation: H.M.C., S.T., S.R., F.-D.C., T.A.V.; drafting the article: H.M.C., S.T., S.R., T.A.V.; critical revision of the article: H.M.C., S.T., S.R., T.A.V.; final approval of the version to be published: H.M.C., S.R., L.W., F.-D.C., L.Z., P.L.C., S.T., T.A.V.

## Competing interests

The authors declare no competing interests.
