## [Peer Review File · Nature Communications]

Reviewer #1 (Remarks to the Author):

In this study, Homeira Moradi Chameh and colleagues characterize the electrophysiological properties of single neurons primarily located in layers 2/3 and 5 of the human anterior temporal lobe. The authors show that sag currents, a functional measure of h channel protein expression, contribute to inter-laminar differences that might have implications for network function and emergent properties and oscillations tied to fundamental aspects of brain function. It is also of interest that the sag currents correlated with patient age, and that these important findings could be replicated in a separate large dataset of human neuron recordings from the Allen Institute. While the study adds to our growing knowledge of human neuron functional properties and is indeed a sizable dataset, it is conspicuously descriptive in nature. That said, such studies are an important contribution to the field, given our limited knowledge of human brain cell type diversity and functional properties, and the rare opportunity to collect this type of data from surgery-derived specimens. Nonetheless, it stands that some methodological considerations (as described in detail below) limit the conclusions and new insights that can be drawn from the present work, and thus, dampen enthusiasm to some extent. In the present form, it is not clear that this work will be of high enough interest to the broad readership of Nature Communications, and it seems a better fit for a specialty journal in neurophysiology or cellular neuroscience. That said, it is possible that additional targeted experiments could be performed to address the concerns and strengthen the findings. It is important to publish rigorous results about human neuron properties if we are to uncover human-specific features of brain cell types and their contributions to health and disease.

Detailed comments:

1) The electrophysiology experiments are well-executed and the analysis of cellular features is sound. However, it seems potentially ill-conceived to treat layers in human neocortex as discrete groupings for aggregating data and comparison of features across layers. In particular, the human L2/3 is greatly expanded in size compared to mammals such as widely studied rodent L2/3, which is of high interest in evolutionary neuroscience and for understanding unique cognitive abilities of humans. A coarse inspection of histological stains or Golgi fills reveals a substantial heterogeneity and clear gradients of cell density, size, and morphologies throughout this region (see also Mohan et al, 2015 Cerebral Cortex; Deitcher et al, 2017 Cerebral Cortex; Kalmbach et al, 2018 Neuron). It is expected that such diversity relates to functional differences, and perhaps to discrete types (as recently described by Hodge et al, 2019 Nature based on gene expression and clustering methods). The worry is that aggregation of data from L2/3 (or 5) masks deeper diversity, and possibly functionally important distinctions that reside within. For example, significant differences in the F/I curves of two types of L2/3 human pyramidal neurons have been described in Deitcher et al, 2017 (not cited in this study), similar to the differences found between L2/3 vs L5 populations in Fig 3c here. Similarly, the differences found for sag and sag ratio for L2/3 vs L5 neurons in this study is of similar magnitude to differences reported for superficial L2 vs deeper L3 human pyramidal neurons in a previous study (Kalmbach et al, 2018). Although the author's note that the detailed position of recorded neurons within L2/3 depth was not recorded or appreciated (and partially addressed with minimal additional data in Fig S1), this stands as a significant down-side limiting the interpretation of the data.

2) Notably, the Beaulieu-Laroche et al 2018 Cell study also described similar values and differences in human L2/3 vs L5 pyramidal neurons, although not presented explicitly as comparisons across lamina (as the focus was on rat vs human comparisons in L5 and L2/3). The relevant data on L2/3 is

easily missed in the supplemental data. Regardless, these findings should be discussed and cited, and perhaps can be used to bolster the findings of the current study. Although the data itself may not be completely novel on human L2/3 and L5 pyramidal neuron electrophysiological properties, the direct comparisons across layers, loose connection to circuit properties and oscillations, and data interpretation in the present study is still novel.

3) The absence of strong evidence for (theta) resonance in either L5 or L2/3 pyramidal neurons from the chirp analysis is further suggestive of potential sampling bias in the dataset that could skew the interpretation of the data. The neuron types most likely to exhibit this cellular property (related to sag amplitude) are either found in the deepest part of L3c of human temporal cortex (discussed in Deitcher et al 2017), or are among the rarest L5 thick-tufted (presumed sub-cortically projecting) type (Beaulieu-Laroche et al, 2019; Hodge et al, 2019). Although admittedly this is not common knowledge from published studies, I mention here with the intent of stimulating further thought on this point, and whether additional targeted sampling (especially the magnopyramidal neurons in deep L3c) might help to clarify on this evidence and whether or not subpopulations in both layers do in fact exhibit strong subthreshold resonance. Targeted experiments are indeed helpful, as demonstrated by Beaulieu-Laroche et al, 2019, where rare thick-tufted L5 pyramidal neurons were recorded by targeting of their prominent apical dendrites, which can be readily distinguished by expert dendritic patchers. Clarification on potential sampling bias and due diligence on searching for these noted types would strengthen the interesting findings in Figs 4 and 5 of the present study.

4) Related to potential sampling bias and position of neurons in the cortical depth, in Fig 6, the correlations and regression lines will be heavily influenced by the exact distribution of depth from pia of the collective cells sampled for each donor. Although this can't be deconvolved at this stage, it should be mentioned as a caveat of the data interpretation.

5) There is not sufficient evidence presented to determine if the lack of subthreshold resonance in L2/3 (especially deep L3c) and L5 neurons might be the consequence of partial truncation of the apical dendrites. This is of interest to address in more detail because it is known that h channels that impart resonance are highly concentrated in the distal dendrites of L5 pyramidal neurons (see Beaulieu-Laroche et al, 2019). Certainly, the high range in the distribution of Rn values may suggest a proportion of cells with truncations, and this is a common and perhaps unavoidable issue in dealing with human pyramidal neurons that have apical dendrites up to 2 mm in length (see also Mohan et al, 2015 Cerebral Cortex). Given that dendritic truncation can also greatly impact bursting behavior of pyramidal neurons, it is probably important to at least mention this point, although clearly this did not impede detection of higher proportion of bursting neurons in human L5 vs L2/3 (but also previously described in Beaulieu-Laroche et al, 2019, which should be noted).

Reviewer #2 (Remarks to the Author):

The authors describe the differences in passive and active membrane properties of L2/3 and L5 pyramidal cells in human brain slices obtained mainly from the anterior temporal lobe of epilepsy patients. They found L5 PCs to have higher input resistance, more h-current, and higher gain in response to noisy inputs at delta to theta frequency range compared to L2/3 PCs. The authors propose that these differences in intrinsic properties may serve as the cellular underpinnings for the generation of theta oscillation at deeper layers and its subsequent propagation to more superficial layers, which they (and others) have reported previously.

The difference in h-current reported here is not surprising given that it is well established from rodent studies that neocortical L5 PCs, particularly the subcortically-projecting subpopulation, have large h-currents, while L2/3 PCs have little to none (van Aerde & Feldmeyer, 2015). This was also confirmed in human studies conducted from samples similar to what was used in the current manuscript (Kalmbach et al., 2018, Beaulieu-Laroche et al., 2018). Contribution of h-current to increased input impedance at the delta-theta frequency range and intrinsic burst firing has likewise been explored previously in both rodent and human studies (Hu et al., 2002; Kalmbach et al., 2018).

A strength of the study lies in its attempt to provide a mechanistic explanation for phenomena that have been observed at a more mesoscopic level *in vivo* in humans. However, the connection is weak, and most of the findings at the cellular level have already been reported previously from both rodent and human samples. The implied causal relationship between the stronger h-current in human L5 PCs and their increased responsiveness to theta input, despite some existing literature making this argument, is not immediately evident in the current manuscript to be conclusive. In addition, there are serious concerns regarding the electrophysiological data, methodology, and analysis.

Major points:

1. The overarching theme throughout the manuscript is that the prominence of h-current in human L5 PCs endows them with increased responsiveness to delta to theta frequency inputs as well as intrinsic bursting properties. However, it is unclear that the h-current is responsible for the generation of theta oscillation by influencing the frequency-dependent gain of each cell type, especially considering that the authors report L5 PCs to have higher input resistance and slower membrane time constant compared with L2/3 PCs. It appears that, as a minimum, repeating the frequency-dependent gain analysis in the presence of ZD7288 would be necessary to support the authors' claim in this context.

2. While the criteria are not agreed upon and the terminology is diverse, it is widely accepted that neocortical L5 pyramidal neurons fall largely into at least two major cell types, with distinguishable long-range projection targets, morphology, and membrane properties, including the size of the h-current (Dembrow et al., 2010). The two different subpopulations also exhibit contrasting local and long-range connectivity, at least in different neocortical regions of rodents (Morishima & Kawaguchi, 2006; Morishima et al., 2011; Morishima et al., 2017; Collins et al., 2018; Anastasiades et al., 2018). Additionally, emerging evidence suggests the presence of cell-type specific properties amongst L2/3 neurons specifically in human slices (Kalmbach et al., 2018), specifically that superficial neurons have little to no h-current and deeper neurons have significant amounts. In light of these points, it is unclear why the authors have decided to group all their recordings into L5 vs L2/3. This is a major potential confound for the authors' interpretation that could be addressed with some straightforward classification and/or clustering.

3. Much of the electrophysiological data presented in the current manuscript are hard to reconcile with previous studies. First, L2/3 PCs were reported to have higher input resistance compared with L5 PCs, in both rodents and in humans (Bitzenhofer et al., 2017; Beaulieu-Laroche et al., 2018; Larkum & Zhu, 2002; Larkum et al., 2007), even though reports in the opposite direction are also present from different authors, albeit not within a single paper (Chang et al., 2005; Luebke et al.,

2007). The discrepancy in terms of input resistance may be, however, due simply to technical differences as different groups tend to have different definitions of input resistance and voltage sag in the presence of h-current. It would therefore be helpful if the authors stated this clearly in their methods and discussion the many substantial disagreements with the published literature. Second, the gain of the current-frequency relationships in Fig. 3C also appear to be higher compared to previous observations, with L2/3 cells already saturating at +350 pA injection. Third, a previous study by Beaulieu-Laroche et al. reported an absence in correlation between patient age and dendritic h-current (albeit with a smaller sample number), which disagrees with one of the major points of the current manuscript. In addition, the correlations the authors report may be subject to methodological and/or statistical problems: despite the massive number of neurons in the Allen Institute dataset for L2/3 neurons, there is essentially no trend, in contrast to the authors' L2/3 data. The authors' L2/3 correlation (Fig. 6C) appears to be strongly driven by a surprising disconnect between the second-oldest patient group in the mid-40s and the oldest patient group of what appears to be early-50s. How is this kind of sampling possible? If the authors left this conspicuous older group out, there would be no correlation across 20 years of patient data. Similarly, the strong correlation in Fig. 6A appears driven by a relatively few neurons from 20-25 year olds with low (even negative!) sag ratios, which seems highly implausible. As does the three 80+ year old patients with extremely high sag ratios in the Allen Institute database. These kinds of correlational analyses are fraught with potential confounds, one of which is likely the sampling of different cell types in the different patients. The authors need to do a much more rigorous and principled job of dealing with these data than just blindly fitting scatterplots.

4. The data in the manuscript exhibits a surprising degree of variance, potentially coming from poor experimental quality control and/or inappropriate analyses. For example, Fig. 1 B-D shows enormous ranges for values that are generally agreed upon to be very tightly controlled in a given cell type. A 20 to 30 mV range in resting potential for pyramidal neurons is completely unprecedented and does not correspond to the vast majority of previous studies. The same goes for input resistance and membrane time constant. The authors must either explain this remarkable variance or cluster/categorize/curate their data according to previously established criteria.

5. References are placed in inappropriate contexts. As an example, the authors reference four papers by Sakmann and Petersen groups in support of their statement that L5 PCs are more excitable than L2/3 cells (page 20). However, all of these papers are exclusively about L2/3 PCs, and none of them mention L5 PCs (except a couple of deeper-positioned cells in the supplement for one of the papers), let alone their input resistances. Furthermore, all of these experiments were conducted *in vivo*, wherein synaptic conductances will reduce the apparent input resistance of the cell (Destexhe et al., 2001), making direct comparisons with *in vitro* studies from slices inappropriate. As another example, the authors point to Larsen et al. (2008) to describe the connection properties of intrinsically bursting cells, but said paper is strictly structural and does not discuss electrophysiology. This issue needs to be addressed.

6. The authors state that they used paired or unpaired t-tests (page 7), except for frequency-dependent gain analysis. The majority of the data, however, are obviously not in normal distribution. Nonparametric tests would therefore be more appropriate.

7. Whole-cell voltage-clamp is not quantitative in neurons with dendrites and voltage-dependent ion channels. It is not clear what the data in G-I add to the manuscript. It should be removed.

Minor points:

1. The depths of the cell bodies for the reconstructed cells in Fig. 1A don't correspond to the anatomical images: are L2/3 and L5 PCs offset from one another, or on different scales? The authors should show laminar borders for these reconstructions if that is how they define L2/3 vs L5. A further anatomical point: are the authors concerned that what they are calling L5 PCs appear to have no apical tuft dendrites?
2. The authors describe that "the half-width of the action potential was shorter in L5 pyramidal cells compared to L2/3 pyramidal cells" (page 13). The associated p-value, however, is 0.07 and should be considered nonsignificant. Either correct the p-value or rephrase the statement.
3. From the solution compositions in methods, it would be more likely for the liquid junction potential to be - 10.8 mV, instead of the + 10.8 mV as appears on the manuscript (page 4).
4. "The membrane time constant (τ_m) was calculated using a single-exponential fit of the membrane potential response to a small hyperpolarizing pulse." (page 4): what was the amplitude of the hyperpolarizing current injection (instead of "small")?
5. " I_h appeared to be one of the major contributors to the prominent post-inhibitory depolarization" (page 20): Hyperpolarizing current injection is not "inhibitory" per se.
6. The definitions of stimulus-response correlation (Csr) and stimulus-stimulus autocorrelation (Css) are not given in the methods (page 6). They are given in the referenced Higgs & Spain paper.
7. Referring to the resonance frequency in the impedance amplitude profile formed by h-current as "low frequency membrane resonance" (page 15) sounds misleading because the effects of the h-current is to impose a high-pass filter. Perhaps rephrasing this could be better.
8. There are numerous typographical errors throughout the manuscript that pose as distractions for the reader. Some, but not all, examples include: Repeated occurrences of "*", "**", "***", indicate significantly different at $P < 0.001$, $P < 0.05$, $P < 0.001$ " (in legends), "L5: $68 \pm 8mV$ " (page 9, describing RMP), "lager sag" (page 10), "we also we performed" (page 11), "we surprisingly we found" (page 15), "Wilcox test" (page 18), " I_{Nap} " (page 20, with undercase p), "Hu et. al" (page 21), and initials of the first author (page 22).

Reviewer #3 (Remarks to the Author):

This is a thorough, careful and thoughtful MS quantifying the distribution of I_h in human – mainly anterior temporal – neocortex by layer. I completely concur with the ethos behind the study – that modelling the human brain demands constraint by human data and, for this reason alone, the paper constitutes a valuable reference work and so to me, deserves publication.

I do have a couple of issues with the MS as it stands though. The comments below are, hopefully, easily dealt with and are intended merely as suggestions to improve the MS.

1) I see why the authors 'hang' their findings on the human theta rhythm. But I don't think its valid

given precedents in rodent literature. As they argue in the discussion, there are multiple mechanisms that can underlie neuronal rhythmicity at theta frequency. Ih is almost certainly not one of them in neocortex. Hippocampal theta cannot reliably be considered the same phenomenon, the frequencies are different for a start. Here there is a dependence on Ih but it lies principally in the behaviour of a subset of interneurons and the neocortical equivalents were not examined in the present MS. In addition, Ih is exquisitely dependent on neuromodulatory state: Theta is mainly seen in non-invasive human recordings in the wake state but, in this condition, multiple wake-associated neuromodulators all act to reduce Ih considerably (e.g. Ach, Orexin/hypocretin etc.). What Ih IS critically involved in is rebound following synaptic inhibition. Thus it plays a crucial role in delayed responses to sensory input (the 'off' response) and 'anodal break' spiking seen in mismatch responses.

2) The data is presented with commendable clarity, but consequently suggests multimodal distributions in a number of the intrinsic cell properties measured. This is discussed briefly (L2 vs L3 for example) but the main subdivision of cell type I know of that manifests in part as sag amplitude differences is between L5a and L5b. The authors have gathered an impressive set of data so I wonder if it is possible to stratify L2/3 and L5 cell types further on the basis of other intrinsic properties (slow AHP, burst generation, afterdepolarisation strength etc.) This may help to 'clean up' the often very broad distributions in some of the metrics.

2 minor points:

3) Lack of observed resonance on somatic recordings with patch electrodes is not surprising. Patch solutions dialyse cytosol hugely and thus interfere with many intrinsic conductances. In addition, the distribution of Ih in neurons is not uniform and resonance can be seen in dendritic recordings in a given cell type when it appears completely absent in somatic recording.

4) The MS data does agree with the Carracedo data in terms of layers showing most theta (discussion). In that paper the field theta was mainly manifest in superficial layers but the origin of the synaptic activity underlying this was exclusively intrinsic theta activity in a subpopulation of L5 cells. See point 1 above though, this theta was not Ih-dependent.

Reviewers' comments:

Reviewer#1 (Remarks to the Author):

In this study, Homeira Moradi Chameh and colleagues characterize the electrophysiological
properties of single neurons primarily located in layers 2/3 and 5 of the human anterior temporal
lobe. The authors show that sag currents, a functional measure of h channel protein expression,
contribute to inter-laminar differences that might have implications for network function and
emergent properties and oscillations tied to fundamental aspects of brain function. It is also of
interest that the sag currents correlated with patient age, and that these important findings
could be replicated in a separate large dataset of human neuron recordings from the Allen
Institute. While the study adds to our growing knowledge of human neuron functional properties
and is indeed a sizable dataset, it is conspicuously descriptive in nature. That said, such studies
are an important contribution to the field, given our limited knowledge of human brain cell type.

Diversity and functional properties; and the rare opportunity to collect this type of data from
surgery-derived specimens. Nonetheless, it stands that some methodological considerations (as
described in detail below) limit the conclusions and new insights that can be drawn from the
present work, and thus, dampen enthusiasm to some extent. In the present form, it is not clear
that this work will be of high enough interest to the broad readership of Nature Communications,
and it seems a better fit for a specialty journal in neurophysiology or cellular neuroscience. That
said, it is possible that additional targeted experiments could be performed to address the
concerns and strengthen the findings. It is important to publish rigorous results about human
neuron properties if we are to uncover human-specific features of brain cell types and their
contributions to health and disease.

Detailed comments:

1) The electrophysiology experiments are well-executed, and the analysis of cellular features is
sound. However, it seems potentially ill-conceived to treat layers in human neocortex as discrete
groupings for aggregating data and comparison of features across layers. In particular, the
human L2/3 is greatly expanded in size compared to mammals such as widely studied rodent
L2/3, which is of high interest in evolutionary neuroscience and for understanding unique
cognitive abilities of humans. A coarse inspection of histological stains or Golgi fills reveals a
substantial heterogeneity and clear gradients of cell density, size, and morphologies throughout
this region (see also Mohan et al, 2015 Cerebral Cortex; Deitcher et al, 2017 Cerebral Cortex;
Kalmbach et al, 2018 Neuron). It is expected that such diversity relates to functional differences,
and perhaps to discrete types (as recently described by Hodge et al, 2019 Nature based on gene
expression and clustering methods). The worry is that aggregation of data from L2/3 (or 5) masks
deeper diversity, and possibly functionally important distinctions that reside within. For
example, significant differences in the F/I curves of two types of L2/3 human pyramidal neurons

have been described in Deitcher et al, 2017 (not cited in this study), similar to the differences
found between L2/3 vs L5 populations in Fig 3c here. Similarly, the differences found for sag and
sag ratio for L2/3 vs L5 neurons in this study is of similar magnitude to differences reported for
superficial L2 vs deeper L3 human pyramidal neurons in a previous study (Kalmbach et al, 2018).
Although the author's note that the detailed position of recorded neurons within L2/3 depth was
not recorded or appreciated (and partially addressed with minimal additional data in Fig S1), this
stands as a significant downside limiting the interpretation of the data.

We thank the reviewer for articulating this concern and note that each of the reviewers raised a
similar issue with our prior submission.

We have addressed this concern in our re-submission:

First, as suggested by this reviewer, we have now conducted a targeted set of recordings of deep
layer 3c pyramidal cells (at 1100-1400 μm distance from pia), allowing us to better parse the
heterogeneity inherent in our prior layer 2/3 pyramidal cell recordings. These recordings allowed
53 us to replicate prior findings from Kalmbach, B et al (2018) showing that sag amplitudes increase
as a function of depth within superficial cortical layers in human cortex (Fig 2B) and that L3c
pyramidal cells are more likely to be resonant (Fig 4).

Second, we now present in Fig 6 an analysis of cell type sub-clusters and gradients based on
multi-variate correlations in pyramidal cells' basic electrophysiology features, such as input
resistance, resting potential, action potential width at rheobase, etc., with an attempt to connect
these gradients to cell morphologies (where available). This analysis suggested that, as described
previously, there exists a great degree of heterogeneity within each of the cellular layers
sampled here, with cells sampled from L2/3 and Layer 3c often displaying similar
electrophysiological features to those from Layer 5 pyramidal cells. This analysis corroborates
the reviewer's comment and considerable previous work suggesting that there is considerable
heterogeneity within cells sampled from each layer (Berg, Sorensen et al. 2020), with much of
this variability being correlated with morphological differences (such as those shown in
(Deitcher, Eyal et al. 2017), and that this is comparable to (or even greater than)
electrophysiological differences across layers.

2) Notably, the Beaulieu-Laroche et al 2018 Cell study also described similar values and
differences in human L2/3 vs L5 pyramidal neurons, although not presented explicitly as
comparisons across lamina (as the focus was on rat vs human comparisons in L5 and L2/3). The
relevant data on L2/3 is easily missed in the supplemental data. Regardless, these findings should
be discussed and cited, and perhaps can be used to bolster the findings of the current study.
Although the data itself may not be completely novel on human L2/3 and L5 pyramidal neuron
electrophysiological properties, the direct comparisons across layers, loose connection to circuit
properties and oscillations, and data interpretation in the present study is still novel.

We sincerely thank the reviewer for highlighting this recent paper that is highly relevant to our
study. While we had cited this paper previously, not including a more thorough comparison of
our data to that from this work was a major oversight on our part in our prior submission. In the
revised manuscript, we have attempted to compare and contrast our findings with those from
(Beaulieu-Laroche, Toloza et al. 2018).

In brief, we feel that our findings are mostly consistent with those reported in Beaulieu-Laroche,
L et al (2018), with the caveat that the data in Beaulieu-Laroche, L et al (2018) were targeted
towards larger, thick-tufted cells in L5, whereas our data from L5 are likely a mix of thin- and
thick-tufted cells (with a suspected bias towards thin-tufted cells). To the extent that there are
differences between our studies, for example, that Beaulieu-Laroche, L et al (2018) report lower
cellular input resistances in L5 relative to L2/3 whereas we do not, our hypothesis is that this
difference in sampling strategies likely underlie such differences.

3) The absence of strong evidence for (theta) resonance in either L5 or L2/3 pyramidal neurons
from the chirp analysis is further suggestive of potential sampling bias in the dataset that could
skew the interpretation of the data. The neuron types most likely to exhibit this cellular property
(related to sag amplitude) are either found in the deepest part of L3c of human temporal cortex
(discussed in Deitcher et al 2017), or are among the rarest L5 thick-tufted (presumed sub-
cortically projecting) type (Beaulieu-Laroche et al, 2019; Hodge et al, 2019). Although admittedly
this is not common knowledge from published studies, I mention here with the intent of
stimulating further thought on this point, and whether additional targeted sampling (especially
the magnopyramidal neurons in deep L3c) might help to clarify on this evidence and whether or
not subpopulations in both layers do in fact exhibit strong subthreshold resonance. Targeted
experiments are indeed helpful, as
demonstrated by Beaulieu-Laroche et al, 2019, where rare thick-tufted L5 pyramidal neurons
were recorded by targeting of their prominent apical dendrites, which can be readily
distinguished by expert dendritic patchers. Clarification on potential sampling bias and due
diligence on searching for these noted types would strengthen the interesting findings in Figs 4
and 5 of the present study.

We thank this reviewer (and a helpful Skype call with Brian Kalmbach) for guidance on how to
better target pyramidal cells that are likely to display subthreshold resonance. In new data
presented within our resubmission, we have now targeted recordings to a subset of pyramidal
cells in human layer 3c. In addition, we have re-analyzed our prior ZAP/chirp data, taking care to
more carefully quantify subthreshold resonance at the “cell-level”, as opposed to the layer level
as in our prior submission.

In our new analysis (Fig 4), we have now found a modest number of cells in each cortical layer
that display subthreshold resonance. We note, however, that the incidence of subthreshold
resonance at “theta” frequencies (> 2Hz) was still quite rare in our dataset, especially compared
to recent reports from Beaulieu-Laroche, L et al (2018) and Kalmbach, B et al (2018). It is certainly
possible that inadvertent confounds, such as dendrite truncation or cytosol dialysis (suggested

by reviewer #3), might obscure the greater expected prevalence of subthreshold resonance in
these data. In the revised manuscript, we have now added text to our discussion (in the
“limitations” subsection) contextualizing our findings and stating these possible sampling and
truncation issues more clearly.

4) Related to potential sampling bias and position of neurons in the cortical depth, in Fig 6, the
correlations and regression lines will be heavily influenced by the exact distribution of depth
from pia of the collective cells sampled for each donor. Although this can't be deconvolved at
this stage, it should be mentioned as a caveat of the data interpretation.

We thank the reviewer for pointing out this important issue of sampling bias and cortical depth
as a potential confound in our analysis of Fig 6 (now Fig 8). We have now included further text
in this section pointing out this confound.

5) There is not sufficient evidence presented to determine if the lack of subthreshold resonance
in L2/3 (especially deep L3c) and L5 neurons might be the consequence of partial truncation of
the apical dendrites. This is of interest to address in more detail because it is known that h
channels that impart resonance are highly concentrated in the distal dendrites of L5 pyramidal
neurons (see Beaulieu-Laroche et al, 2019). Certainly, the high range in the distribution of R_n
values may suggest a proportion of cells with truncations, and this is a common and perhaps
unavoidable issue in dealing with human pyramidal neurons that have apical dendrites up to 2
137 mm in length (see also Mohan et al, 2015 Cerebral Cortex). Given that dendritic truncation can
also greatly impact bursting behavior of pyramidal neurons, it is probably important to at least
mention this point, although clearly this did not impede detection of higher proportion of
bursting neurons in human L5 vs L2/3 (but also previously described in Beaulieu-Laroche et al,
2019, which should be noted).

This is a fair point. It's certainly possible that the lack of observation of more subthreshold
resonance is impacted by inadvertent truncation of dendrites. The high range in cell input
resistances reported here would also support the observation that some of this range is likely
due to truncation (see new supplemental figure S7 for further corroboration on this point using
data from the Allen Institute where apical dendrite truncation was more systematically
annotated). We have now taken care to mention the confound of dendrite truncation when
reporting our results regarding input resistances and resonance.

Regarding the point about bursting, we have re-analyzed our data on bursting to use a definition
of bursting that is inspired by that used in Beaulieu-Laroche, L et al (2018) (i.e., based on the
instantaneous firing rate at or near rheobase). As reported previously by Beaulieu-Laroche, L et
al (2018), we similarly see a relatively low incidence of bursting among cells from each major
layer within our dataset.

Reviewer #2(Remarks to the Author):

The authors describe the differences in passive and active membrane properties of L2/3 and L5
pyramidal cells in human brain slices obtained mainly from the anterior temporal lobe of
epilepsy patients. They found L5 PCs to have higher input resistance, more h-current, and higher
gain in response to noisy inputs at delta to theta frequency range compared to L2/3 PCs. The
authors propose that these differences in intrinsic properties may serve as the cellular
underpinnings for the generation of theta oscillation at deeper layers and its subsequent
propagation to more superficial layers, which they (and others) have reported previously.

The difference in h-current reported here is not surprising given that it is well established from
rodent studies that neocortical L5 PCs, particularly the subcortically-projecting subpopulation,
have large h-currents, while L2/3 PCs have little to none (van Aerde & Feldmeyer, 2015). This
was also confirmed in human studies conducted from samples similar to what was used in the
current manuscript (Kalmbach et al., 2018, Beaulieu-Laroche et al., 2018). Contribution of h-
current to increased input impedance at the delta-theta frequency range and intrinsic burst
firing has likewise been explored previously in both rodent and human studies (Hu et al., 2002;
Kalmbach et al., 2018). A strength of the study lies in its attempt to provide a mechanistic
explanation for phenomena that have been observed at a more mesoscopic level in vivo in
humans. However, the connection is weak, and most of the findings at the cellular level have
already been reported previously from both rodent and human samples. The implied causal
relationship between the stronger h-current in human L5 PCs and their increased responsiveness
to theta input, despite some existing literature making this argument, is not immediately evident
in the current manuscript to be conclusive. In addition, there are serious concerns regarding the
electrophysiological data, methodology, and analysis.

Major points:

1. The overarching theme throughout the manuscript is that the prominence of h-current in
human L5 PCs endows them with increased responsiveness to delta to theta frequency inputs as
well as intrinsic bursting properties. However, it is unclear that the h-current is responsible for
the generation of theta oscillation by influencing the frequency-dependent gain of each cell type,
especially considering that the authors report L5 PCs to have higher input resistance and slower
membrane time constant compared with L2/3 PCs. It appears that, as a minimum, repeating the
frequency-dependent gain analysis in the presence of ZD7288 would be necessary to support
the authors' claim in this context.

As suggested, we have now repeated the frequency dependent gain analysis experiments in the
presence of ZD7288. We found that blocking \$I_h\$ predominantly abolished the delta peak in L5
cells, and to a lesser extent the theta peak in both L2/3 and L5 cells. These data provide further

evidence that the I_h in L5 pyramidal cells imbues them with increased responsiveness to delta
and theta frequency input.

We agree with the reviewer that h-current is but one of many mechanisms for generating theta
oscillations, with a number of other candidate mechanisms also possible. We have highlighted a
number of these alternative mechanisms in our discussion.

2. While the criteria are not agreed upon and the terminology is diverse, it is widely accepted
that neocortical L5 pyramidal neurons fall largely into at least two major cell types, with
distinguishable long-range projection targets, morphology, and membrane properties, including
the size of the h-current (Dembrow et al., 2010). The two different subpopulations also exhibit
contrasting local and long-range connectivity, at least in different neocortical regions of rodents
(Morishima & Kawaguchi, 2006; Morishima et al., 2011; Morishima et al., 2017; Collins et al.,
2018; Anastasiades et al., 2018). Additionally, emerging evidence suggests the presence of cell-
type specific properties amongst L2/3 neurons specifically in human slices (Kalmbach et al.,
2018), specifically that superficial neurons have little to no h-current and deeper neurons have
significant amounts. In light of these points, it is unclear why the authors have decided to group
all their recordings into L5 vs L2/3. This is a major potential confound for the authors'
interpretation that could be addressed with some straightforward classification and/or
clustering.

We acknowledge this important point that it is extremely likely that there are subtypes within
each of the major cortical layer-based groupings that we have presented here.

As suggested by this reviewer and reviewer #1, a major limitation with our prior analysis was our
somewhat naïve grouping of pyramidal cells into layers. To address this comment, we have
conducted a number of additional targeted recordings from Layer 3c, with the goal of confirming
the finding that there is a greater degree of sag / h-current in the deeper parts of Layer 3c relative
to more superficial cortex.

In addition, we now present in Fig 6 an analysis of cell type sub-clusters and gradients based on
multi-variate correlations in pyramidal cells' basic electrophysiology features, such as input
resistance, resting potential, action potential width at rheobase, etc, with an attempt to connect
these gradients to cell morphologies (where available). This analysis suggested that, as described
previously, there exists a great degree of heterogeneity within each of the cellular layers
sampled here, with cells sampled from Layers 2/3 and 3c often displaying similar
electrophysiological features to those from Layer 5 pyramidal cells. Regarding the point about
multiple cell types being present, in particular thin- and thick-tufted cells within our data from
human L5, we feel that this suspicion is more or less confirmed through a qualitative comparison
of our data with the limited reconstructions that were available. However, we note that we were
not able to easily parse L5 ET from IT cells from our data with this approach, just that we could
likely confirm that such a dichotomy likely is present within our data.

This analysis corroborates the reviewer's point and considerable previous work suggesting that
there is considerable heterogeneity within cells sampled from each layer (Berg, Sorensen et al.
2020), with much of this variability being correlated with morphological differences (such as
those shown in (Deitcher, Eyal et al. 2017), and that this comparable to, or even greater than,
electrophysiological differences across layers.

3a. Much of the electrophysiological data presented in the current manuscript are hard to
reconcile with previous studies. First, L2/3 PCs were reported to have higher input resistance
compared with L5 PCs, in both rodents and in humans (Bitzenhofer et al., 2017; Beaulieu-Laroche
et al., 2018; Larkum & Zhu, 2002; Larkum et al., 2007), even though reports in the opposite
direction are also present from different authors, albeit not within a single paper (Chang et al.,
2005; Luebke et al., 2007). The discrepancy in terms of input resistance may be, however, due
simply to technical differences as different groups tend to have different definitions of input
resistance and voltage sag in the presence of h-current. It would therefore be helpful if the
authors stated this clearly in their methods and discussion the many substantial disagreements
with the published literature.

We agree that it is surprising that we find that L5 pyramidal cells (as a group) have similar input
resistances to L2/3 pyramidal cells (again, as a group). We don't feel that this finding is due to
mere differences in how input resistance is calculated, as we obtained similar estimates of input
resistances using multiple methods (e.g., manually calculating them in ClampFit, calculating
them algorithmically in Python). We acknowledge that it is certainly possible that the input
resistances we report here are biased upwards here due to inadvertent cutting of dendrites,
especially for larger cells, like those in L5, we have now added a new section to the discussion to
further describe this limitation.

We contextualize our finding of higher (or similar) group-averaged input resistances between L5
and L2/3 pyramidal cells as follows: it is very likely that our dataset is oversampled for thin-tufted
compared to thick-tufted cells, especially compared to the L5 pyramidal cells sampled by
Beaulieu-Laroche, L et al (2018), which intentionally targeted larger thick-tufted cells, whereas
our sampling of cells in L5 tended to be more random. Based on recent single-cell transcriptomics
data from human neocortex by Allen Institute that further compared these cells' transcriptomes
to orthologous cell types in the mouse, there is strong (transcriptomic) evidence that extra-
telencephalic (thick-tufted) cells comprise roughly 1% of the excitatory cells in MTG in human
L5, as opposed to approximately 20% in the rodent. Therefore, it is quite likely that most of our
(randomly-sampled) cells from L5 are likely to be IT / thin-tufted, and if so, would explain the
discrepancy between the relatively large input resistances reported here versus those sampled
in Beaulieu-Laroche, L et al (2018) or in previous work in rodents. We note that without
additional morphologies or, ideally, molecular corroboration (like with Patch-seq), we cannot
conclusively confirm this hypothesis.

To further corroborate this point, we note that when we have re-analyzed the publicly available
intrinsic electrophysiology data collected from human brain samples from the Allen Institute, we
find a similar trend for L5 cells to show larger input resistances than those from L2 and L3 in their
cohort and that this trend holds even for cells where the apical dendrite has been annotated to
be explicitly intact (Figure S7).

3b. Second, the gain of the current-frequency relationships in Fig. 3C also appear to be higher
compared to previous observations, with L2/3 cells already saturating at +350 pA injection.

We thank the reviewer for their comments regarding the current-frequency relationships (FI
curves) shown in our prior submission.

A detailed reanalysis of our entire dataset (see reviewer comment and response in point 4
below) found that some putative fast-spiking interneurons were included in the dataset in our
prior submission and thus erroneously included in our prior analysis. These outliers led to the
increased gain in the apparent FI curve. In our revised submission (after discarding these putative
interneurons), the updated FI curves now correspond much more closely with our expectations
and those from the previous literature.

We have also included a cursory analysis of the electrophysiological properties of our suspected
interneurons in Figure 5.

3c. Third, a previous study by Beaulieu-Laroche et al. reported an absence in correlation between
patient age and dendritic h-current (albeit with a smaller sample number), which disagrees with
one of the major points of the current manuscript. In addition, the correlations the authors
report may be subject to methodological and/or statistical problems: despite the massive
number of neurons in the Allen Institute dataset for L2/3 neurons, there is essentially no trend,
in contrast to the authors' L2/3 data. The authors' L2/3 correlation (Fig. 6C) appears to be
strongly driven by a surprising disconnect between the second-oldest patient group in the mid-
40s and the oldest patient group of what appears to be early-50s. How is this kind of sampling
possible? If the authors left this conspicuous older group out, there would be no correlation
across 20 years of patient data. Similarly, the strong correlation in Fig.6A appears driven by a
relatively few neurons from 20-25 year olds with low (even negative!) sag ratios, which seems
highly implausible. As does the three 80+ year old patients with extremely high sag ratios in the
Allen Institute database. These kinds of correlational analyses are fraught with potential
confounds, one of which is likely the sampling of different cell types in the different patients.
The authors need to do a much more rigorous and principled job of dealing with these data than
just blindly fitting scatterplots.

We acknowledge each of these very valid concerns with this prior analysis and are grateful for
the note that a similar analysis was performed in Beaulieu-Laroche, L et al (2018) with no
relationship found between dendritic sag and patient age. The comment that we are likely

comparing across different cell types (e.g., IT vs ET cells) in different patients is important.
Similarly, following our extensive data re-curation and re-analysis (which addresses the issue of
negative sag ratios), the p-value of our primary correlation relating increased sag ratio to age in
L5 pyramidal cells is now no longer significant ($p = 0.10$ as opposed to $p = 0.03$ previously).
However, the evidence (albeit weak) for replication (in L5) from data from the Allen Institute
cohort is unchanged.

In light of this mixed-bag of results, we have elected to de-emphasize the results of the
correlational analysis, specifically, removing the mention of this result from the paper title and
the abstract, and using clearer language to better indicate how provisional and weak this
putative relationship is. We have also included a reference Beaulieu-Laroche, L et al (2018).
illustrating that this relationship did not in fact replicate in that dataset.

However, we do not wish to hang the entire manuscript on this point and would be satisfied to
move this analysis to the supplement (or remove it completely from the manuscript all-
together).

4. The data in the manuscript exhibits a surprising degree of variance, potentially coming from
poor experimental quality control and/or inappropriate analyses. For example, Fig. 1 B-D shows
enormous ranges for values that are generally agreed upon to be very tightly controlled in a
given cell type. A 20 to 30 mV range in resting potential for pyramidal neurons is completely
unprecedented and does correspond to the vast majority of previous studies. The same goes for
input resistance and membrane time constant. The authors must either explain this remarkable
variance or cluster/categorize/curate their data according to previously established criteria.

We sincerely thank the reviewer for these careful observations. Guided by these comments, we
have conducted a systematic inspection and curation effort of our entire dataset.

During this re-curation, we discovered two major issues that have now been corrected in our
updated manuscript. First, in our prior analysis, we were inadvertently including data from a
number of putative interneurons (identified by their spike width, maximal firing rates, and spike
after-hyperpolarization amplitude, and for one cell, a morphology). We have now excluded these
putative interneurons from our primary analysis and report them in a separate analysis in Fig 5.
After removal, we now see Firing rate vs injected current curves (FI curves) more in line with one
would expect (Fig 3).

Second, during our re-curation, we noticed that we were mistakenly including a subset of
intrinsic electrophysiology data from some cells where sweeps were initially hyperpolarized to
near -80mV via negative current injection, resulting in an incorrect annotation of these cells'
resting membrane potentials. The inadvertent inclusion of this data contributed to the very large
range (and almost bimodal distribution) of resting potential values in our prior submission. We
have now excluded these traces (replacing them with the correct traces recorded with no current

injection at baseline from these same cells). This correction now addresses some of the
concerning large range in RMPs that were raised by this reviewer (Fig 1).

Lastly, we acknowledge that despite this careful curation effort, these data remain quite
heterogeneous, especially given what one might expect based on analogous studies in the
rodent. Some of this heterogeneity is doubtless due to technical factors given the limitations of
the human surgical tissue that are especially challenging to control (e.g., tissue quality).
However, as suggested by this reviewer and the other reviewers, some of this heterogeneity is
also very likely to be biological, reflecting natural gradients or subclusters within these data
(explored in Fig 6 and discussed in detail in the Discussion).

5. References are placed in inappropriate contexts. As an example, the authors reference four
papers by Sakmann and Petersen groups in support of their statement that L5 PCs are more
excitable than L2/3 cells (page 20). However, all of these papers are exclusively about L2/3 PCs,
and none of them mention L5 PCs (except a couple of deeper-positioned cells in the supplement
for one of the papers), let alone their input resistances. Furthermore, all of these experiments
were conducted in vivo, wherein synaptic conductances will reduce the apparent input
resistance of the cell (Destexhe et al., 2001), making direct comparisons with in vitro studies
from slices inappropriate. As another example, the authors point to Larsen et al. (2008) to
describe the connection properties of intrinsically bursting cells, but said paper is strictly
structural and does not discuss electrophysiology. This issue needs to be addressed.

Thank you for pointing this out. We have now removed these erroneous citations in the revision.

6. The authors state that they used paired or unpaired t-tests (page 7), except for frequency-
dependent gain analysis. The majority of the data, however, are obviously not in normal
distribution. Nonparametric tests would therefore be more appropriate.

We have replaced our previous statistical tests with non-parametric Wilcoxon tests.

7. Whole-cell voltage-clamp is not quantitative in neurons with dendrites and voltage-dependent
ion channels. It is not clear what the data in G-I add to the manuscript. It should be removed.

We appreciate the reviewer's comment regarding the well-known space clamp issues that limit
our ability to adequately clamp these cells' voltage. We acknowledge that these issues would be
especially pronounced for human pyramidal cells given their large size.

Despite these known limitations, we thought that there would be some value in attempting to
use voltage-clamp to help provide kinetics and voltage dependency characteristics for the
human I_h . In light of the concerns raised by the reviewer, we have now elected to 1) move these
data to the supplemental figures (Figure S2); and 2) to address these space clamp issues in the
manuscript text to allow the reader to better contextualize our findings given these limitations.

Minor points:

1. The depths of the cell bodies for the reconstructed cells in Fig. 1A don't correspond to the
anatomical images: are L2/3 and L5 PCs offset from one another, or on different scales? The
authors should show laminar borders for these reconstructions if that is how they define L2/3 vs
L5. A further anatomical point: are the authors concerned that what they are calling L5 PCs
appear to have no apical tuft dendrites?

We have now updated the presentation of the cell morphologies in Fig1A to address this point
of clarity.

2. The authors describe that "the half-width of the action potential was shorter in L5
pyramidal cells compared to L2/3 pyramidal cells" (page 13). The associated p-value,
however, is 0.07 and should be considered nonsignificant. Either correct the p-value or
rephrase the statement.

Corrected in the text.

3. From the solution compositions in methods, it would be more likely for the liquid junction
potential to be - 10.8 mV, instead of the + 10.8 mV as appears on the manuscript (page 4).

Corrected in the text.

4. "The membrane time constant (τ_m) was calculated using a single-exponential fit of the
membrane potential response to a small hyperpolarizing pulse." (page 4): what was the
amplitude of the hyperpolarizing current injection (instead of "small")?

The methodology for calculating membrane time constant has now been more explicitly
described.

5. " I_h appeared to be one of the major contributors to the prominent post-inhibitory
depolarization" (page 20): Hyperpolarizing current injection is not "inhibitory" per se.

Corrected in the text.

6. The definitions of stimulus-response correlation (C_{sr}) and stimulus-stimulus autocorrelation
(C_{ss}) are not given in the methods (page 6). They are given in the referenced Higgs & Spain paper.

We now include the definitions of these quantities.

7. Referring to the resonance frequency in the impedance amplitude profile formed by h-current

as “low frequency membrane resonance” (page 15) sounds misleading because the effects of
the h-current is to impose a high-pass filter. Perhaps rephrasing this could be better.

Corrected in the text.

8. There are numerous typographical errors throughout the manuscript that pose as distractions
for the reader. Some, but not all, examples include: Repeated occurrences of “*, **,***”, indicate
significantly different at $P < 0.001$, $P < 0.05$, $P < 0.001$ ” (in legends), “L5: $68 \pm 8\text{mV}$ ” (page 9,
describing RMP), “lager sag” (page 10), “we also we performed” (page 11), “we surprisingly we
found” (page 15), “Wilcox test” (page 18), “I_Nap” (page 20, with undercase p), “Hu et. al” (page
21), and initials of the first author (page 22).

Corrected in the text

Reviewer #3 (Remarks to the Author):

This is a thorough, careful and thoughtful MS quantifying the distribution of Ih in human – mainly
anterior temporal – neocortex by layer. I completely concur with the ethos behind the study –
that modelling the human brain demands constraint by human data and, for this reason alone,
the paper constitutes a valuable reference work and so to me, deserves publication.
I do have a couple of issues with the MS as it stands though. The comments below are, hopefully,
easily dealt with and are intended merely as suggestions to improve the MS.

1) I see why the authors ‘hang’ their findings on the human theta rhythm. But I don’t think its
valid given precedents in rodent literature. As they argue in the discussion, there are multiple
mechanisms that can underlie neuronal rhythmicity at theta frequency. Ih is almost certainly not
one of them in neocortex. Hippocampal theta cannot reliably be considered the same
phenomenon, the frequencies are different for a start. Here there is a dependence on Ih but it
lies principally in the behaviour of a subset of interneurons and the neocortical equivalents were
not examined in the present MS. In addition, Ih is exquisitely dependent on neuromodulatory
state: Theta is mainly seen in non-invasive human recordings in the wake state but, in this
condition, multiple wake-associated neuromodulators all act to reduce Ih considerably (e.g. Ach,
Orexin/hypocretin etc.). What Ih IS critically involved in is rebound following synaptic inhibition.
Thus it plays a crucial role in delayed
responses to sensory input (the ‘off’ response) and ‘anodal break’ spiking seen in mismatch
responses.

We thank the reviewer for this thorough and positive comment. We indeed agree with the
reviewer’s analysis that theta rhythmicity is a complex dynamic, both at the cellular and network
level, that is driven by a wide variety of factors beyond \$I_h\$, and that there are important

differences between human and rodent theta rhythms. We have thus endeavored to downplay
the emphasis on theta resonance. Our ZD-7288 (I_h blocker) experiments firmly establish that the
low frequency peaks in $G(f)$ particularly in the delta frequency are dependent on I_h . In light of
these new results, we interpret our previous findings of interlaminar coherence at theta
frequency (4-8Hz), as arising from the dynamic interaction between IB cells (that burst at delta),
and RS cells (which are the predominant cell-type we likely recorded from) that receive delta
frequency input from IB cells to which they are particularly tuned to (peak in delay in $G(f)$), which
then discharge at theta frequency (and possibly in part the theta frequency peak in $G(f)$). Please
see response to last comment for a continuation of this discussion.

2) The data is presented with commendable clarity, but consequently suggests multimodal
distributions in a number of the intrinsic cell properties measured. This is discussed briefly (L2 vs
L3 for example) but the main subdivision of cell type I know of that manifests in part as sag
amplitude differences is between L5a and L5b. The authors have gathered an impressive set of
data so I wonder if it is possible to stratify L2/3 and L5 cell types further on the basis of other
intrinsic properties (slow AHP, burst generation, afterdepolarisation strength etc.) This may help
to 'clean up' the often very broad distributions in some of the metrics.

We thank the reviewer for this excellent comment. We point the reviewer to our analysis in (Fig
6), which attempts to address some of the large degree in heterogeneity within cells from the
same cortical layer.

We also note that we address this topic in our responses to similar comments from the other
reviewers: specifically, we have addressed some of this increased variability by identifying
interneurons and distinct experimental protocols that were mistakenly included in our data set,
while noting that some level of increased heterogeneity in these cells (relative to the rodent
setting) is to be expected based on the emerging human cortical cell typing literature.

2 minor points:

3) Lack of observed resonance on somatic recordings with patch electrodes is not surprising.
Patch solutions dialyse cytosol hugely and thus interfere with many intrinsic conductances. In
addition, the distribution of I_h in neurons is not uniform and resonance can be seen in dendritic
recordings in a given cell type when it appears completely absent in somatic recording.

We thank the reviewer for noting these potential confounds in our subthreshold resonance data.
We have now added a paragraph to our discussion within our limitations section to contextualize
how experimental confounds, such as dendrite cutting or cytosol dialysis, might influence some
of our ability to observe subthreshold resonance in these data.

4) The MS data does agree with the Carracedo data in terms of layers showing most theta
(discussion). In that paper the field theta was mainly manifest in superficial layers but the origin

of the synaptic activity underlying this was exclusively intrinsic theta activity in a subpopulation
of L5 cells. See point 1 above though, this theta was not I_h -dependent.

We thank reviewer for this excellent comment and insights. We fully agree that I_h is likely not
directly responsible for the theta frequency activity in local circuitry. We do however argue in
line with the Carracedo, L et al (2013) work, that it is indirectly complicit in theta generation
given our new experiments added to this MS, that show that the delta peak ($\sim 2\text{Hz}$) in $G(f)$, and
less so the theta peak ($\sim 7\text{Hz}$) are dependent on I_h (see figure 7E-F). From the Carracedo, L et al
(2013) paper it was the RS cells that generated theta, likely the cell-type that primarily
contributed to $G(f)$ (Figure 7B). We speculate that they did not observe theta in the deep layers,
since human circuits appear to amplify local activity relative to rodent cortex (Molnár, Oláh et
al. 2008) which might make theta more prominent human L5. Conversely, we did not observe
delta activity in our previous in-vitro recordings, possibly due to the paucity of IB cells in human
middle temporal gyrus cortex (Hodge, Bakken et al. 2019). Our data as well provides a putative
cellular mechanism (peaks in $G(f)$) underlying these population activities. We have schematized
these ideas below and include it as a supplementary (Figure S8) and clarified this in the
discussion.

**Figure S8: Structural circuit motif for L5 theta oscillations.** Delta frequency output from intrinsically
bursting (IB) (Carracedo, Kjeldsen et al. 2013) neurons is well tracked by regular spiking (RS) cells that
have a peak in $G(f)$ (red) (figure 7B in main text) at delta. In turn RS cells that are poorly adapting, have
steep f-I curves, and low rheobase discharge at theta frequency (Carracedo, Kjeldsen et al. 2013). RS cell
drive local circuits at theta including other RS cells that track theta well via the peak at 7Hz in $G(f)$.
Interneurons amplify local activity through rebound excitation (I ; orange; see Figure 5 main text) in
human circuits that are predisposed to reverberant activity (Molnár, Oláh et al. 2008).

**References cited**

- Beaulieu-Laroche, L., E. H. Toloza, M.-S. van der Goes, M. Lafourcade, D. Barnagian, Z. M. Williams, E. N.
Eskandar, M. P. Frosch, S. S. Cash and M. T. Harnett (2018). "Enhanced dendritic compartmentalization
in human cortical neurons." Cell **175**(3): 643-651. e614.
- Berg, J., S. A. Sorensen, J. T. Ting, J. A. Miller, T. Chartrand, A. Buchin, T. E. Bakken, A. Budzillo, N. Dee
and S.-L. Ding (2020). "Human cortical expansion involves diversification and specialization of
supragranular intratelencephalic-projecting neurons." BioRxiv.
- Carracedo, L. M., H. Kjeldsen, L. Cunnington, A. Jenkins, I. Schofield, M. O. Cunningham, C. H. Davies, R.
D. Traub and M. A. Whittington (2013). "A neocortical delta rhythm facilitates reciprocal interlaminar
interactions via nested theta rhythms." Journal of Neuroscience **33**(26): 10750-10761.
- Deitcher, Y., G. Eyal, L. Kanari, M. B. Verhoog, G. A. Atenekeng Kahou, H. D. Mansvelder, C. P. De Kock
and I. Segev (2017). "Comprehensive morpho-electrotonic analysis shows 2 distinct classes of L2 and L3
pyramidal neurons in human temporal cortex." Cerebral Cortex **27**(11): 5398-5414.
- Hodge, R. D., T. E. Bakken, J. A. Miller, K. A. Smith, E. R. Barkan, L. T. Graybuck, J. L. Close, B. Long, N.
Johansen and O. Penn (2019). "Conserved cell types with divergent features in human versus mouse
cortex." Nature **573**(7772): 61-68.
- Molnár, G., S. Oláh, G. Komlósi, M. Füle, J. Szabadics, C. Varga, P. Barzó and G. Tamás (2008). "Complex
events initiated by individual spikes in the human cerebral cortex." PLoS biology **6**(9): e222.

Reviewer #1 (Remarks to the Author):

The authors have done a nice job of being responsive and open to the reviewer concerns and addressing the major issues raised. I appreciate the willingness of the authors to acknowledge various flaws that were either directly called out or discovered on their own in the course of revision. The corrections and new experimental data collected were quite important and no doubt have strengthened the manuscript over the original submission. I acknowledge that quite a lot of new experimental and analysis work was completed for this revision. Given the substantial new material introduced, I have additional comments on those changes, and these new comments are intended to help further strengthen the manuscript. I believe the manuscript could be acceptable for publication in Nature Communications pending these additional relatively minor changes.

Conducting additional experiments on L3c pyramidal neurons was a reasonable compromise to get at gradients of features for supragranular pyramidal neuron population. This was an acceptable choice given that targeting of rare L5 thick tufted neurons in the human cortex would be exceptionally challenging. The conclusions are limited by the lack of L5 ET neurons in the data set, but this caveat is reasonably well discussed and acknowledged), and I find that acceptable.

Regarding L2/3 vs L3c, I worry that this naming scheme may cause confusion for readers familiar with rodent work on L2/3, as there are in fact discrete layers 2 and 3 anatomically in the human cortex. The authors should consider whether this can be better solved in the presentation of the data.

Line 483-484 should be corrected for grammar, e.g. "These results suggest that depth from pia appears to be a general organizing principle for sag current in the cortex for both excitatory and inhibitory neurons." I'm not so sure organizing principle is the correct way to pitch this observation. It is an interesting observation, and is the only real justification to keep this as a main figure. It is a little odd to include this where interneurons in L2/3 and L5 are called "putative" and no information on the diversity of interneuron types is readily available for these data, especially given that the rest of the paper is focused on excitatory neurons across layer. To me this seems to be a supplemental figure at best, and still feels preliminary. The principle result of sag currents being larger in deeper layers relative to upper layers may be a real phenomenon, or it may be a result of interneuron subtype sampling bias in this small interneuron dataset. The authors should consider this issue and at least acknowledge the limitations more directly in the text.

Regarding Figure S4, are these two neuron morphologies shown at matched scale? Likely they are not. It is a peculiar non-pyramidal neuron morphology indeed.

Figure S7: It is striking that the input resistance ranges are completely overlapping between Krembil and Allen data for L2 and L3 pyr neurons, yet the distributions are nearly non overlapping for L5 neurons, which can't be fully accounted for by truncation status. I imagine this discrepancy can't be resolved in the present study, yet it is notable. Despite this fact, it is commendable that the authors utilized this publicly available data set to compare to their data set and attempt to corroborate several findings.

The novel visualization in Fig 6 is interesting but provides more questions than discrete answers. The recent transcriptomics data strongly suggests that layer membership is not particularly informative regarding cell type diversity in human cortex, as many discrete cell types do not obey strict laminar

boundaries. It might be worth expounding a little more on the point that layer is not a good surrogate for discrete cell types.

Reviewer #2 (Remarks to the Author):

For this revised manuscript, the authors report five main findings from their study on human neocortical pyramidal cells: 1) a gradient of increasing intrinsic excitability and Ih from superficial to deeper layers; 2) Ih-dependent enhancement of neuronal gain at delta and theta frequencies for layer 5 cells; 3) subthreshold resonance found in pyramidal cells; 4) heterogeneity in membrane properties; and 5) a possible correlation of Ih with patient age.

Most of these results are not new, having been previously investigated quite deeply, as was mentioned in the preceding round of review. The contribution of Ih to membrane resonance and its potential link with neural oscillation has been known for decades (Hutcheon et al., 1996). The positive correlation between Ih and somatic depth of human layer 2/3 pyramidal cells as well its contribution to frequency-dependent power were also recently described in detail (Kalmbach et al., 2018). Stronger Ih in thick tufted human layer 5 pyramidal cells compared with layer 2/3, consistent with many previous observations from rodents, has likewise been documented (Beaulieu-Laroche et al., 2018). A large proportion of remaining data in the current manuscript, such as action potential waveform analysis, is potentially interesting, but seems detached from the main direction of the study.

The authors emphasize two findings in particular as important: direct experimental demonstration of the consequence of Ih block on frequency-dependent gain in layer 2/3 and 5 pyramidal cells (which was suggested in the previous round of review) and the potential correlation between Ih and age. However, these results are both not surprising based on previous knowledge and inadequately supported by the evidence provided. Additionally, serious concerns persist regarding methodology, data curation, and interpretation; this accrues with inconsistencies between the current study and the existing literature, calling in to question what descriptive value remains.

Major points:

1. The large majority of cells are shown to be non-resonant regardless of their position, with fewer cells displaying very low resonance frequency typically around 1 Hz even though representative examples are taken from the extremes (Fig. 4C). In contrast, Kalmbach et al. report the majority of layer 2/3 cells to be at least modestly resonant, especially in but not limited to the deepest layer 3 cells (Fig. 4B & D of Kalmbach et al.), along with considerably higher resonance frequency compared with values in the current manuscript. While the possible cause of inconsistency between these data is not discussed, the authors address this lack of subthreshold resonance by noting that there are other known contributors to membrane resonance. Unlike Kalmbach et al., such small fraction of resonant cells with very low resonance frequencies do not appear to be in good alignment with, or at least immediately relevant to, one of the central arguments of the manuscript, that Ih shapes neuronal resonance properties in such way that results in enhanced input-output relationship at delta-theta range for layer 5 cells (Fig. 7F). It could be argued that the difference may originate from the subthreshold vs. suprathreshold nature of the stimulation protocols used; even so, the frequency-dependent gain profile from noisy current input (Fig. 7B) is not consistent with that from sinusoidal current input either (Fig. 4F), where the significant and roughly twofold difference in spike probability at higher frequencies as shown in the latter disappears from the former. More apparent discrepancies can be found from Fig. 7, wherein the frequency-dependent gain profiles in panels B vs. E & F show striking difference in tendency which in some ways even appear to be reversed,

particularly for layer 5 cells. It is surprising in this regard that, despite such large variances in data, that the authors felt showing $n = 3$ cells without any statistical analysis was appropriate. I don't even see any error bars. I do not find this experiment convincing, particularly because it's not clear how these few cells were chosen from the heterogeneous population shown in Fig 4.

2. It is remarkable that such cells as cell f or cell g shown in Fig. 1, which only reach up to layer 3 or even the border between layer 3 and 4, are presented as representative examples of layer 5 pyramidal cells that are presumed to be intact along the longitudinal axis. Cell g is further proposed to be a thick tufted layer 5 pyramidal cell (line 539), even though its morphological reconstruction simply lacks a tuft. Such features are completely unencountered in rodents and, to our knowledge, in humans as well. Note that the spiny neurons that are labeled as intact in the Allen Institute database whose electrophysiology and morphology are available (of which there are only two cells) also have visible apical dendrites reaching all the way up to layer 1, while at the same time displaying classic morphological features of neocortical layer 5 pyramidal cells. In addition, it is worthwhile to note here that the presence of spines alone is not a guarantee for a cell to be pyramidal or even excitatory, and interneurons including the sparsely spiny interneurons are also found more frequently in primates than in rodents (Kawaguchi & Kubota, 1993; DeFelipe, 2011). Whether the apical dendrites of cells recorded are intact is of much importance for the current study, not only as it will affect passive membrane properties but more importantly because ion channels including HCN channels are known to be expressed in steep gradients along the somatodendritic axis (Lorincz et al., 2002). It is also unclear from the manuscript how the authors excluded the non-fast-spiking interneurons (which coincidentally have high input resistance) from analysis, which are much less obviously distinguishable from pyramidal cells by electrophysiology alone to the untrained eye. The authors describe that putative interneurons were identified by spike properties such as action potential width, maximum firing rate, and strong afterhyperpolarization, all of which are well-suited criteria for fast-spiking interneurons such as those shown as representative examples, but much less effective for identifying non-fast-spiking interneurons (a population which also overlaps with the aforementioned sparsely spiny interneurons such as calbindin-expressing interneurons in the deeper layers of the neocortex).

3. The authors continue to place irrelevant and inappropriate references in attempts to support their otherwise unfounded claims, even after having been corrected for numerous such instances from their previous manuscript. For example, they state that "human neocortical circuits demonstrate [...] different short-term plasticity rules, compared to neocortical circuits in rodents"; setting aside the ambiguity on exactly what type of synapses the authors are referring to here, one of the two cited papers is entirely on rhythmic firing in rat layer 5 pyramidal cells, while the other is about timing rules for spike timing dependent plasticity in human hippocampal (not neocortical) neurons. Neither study addresses or even mentions short-term plasticity. Another example is their statement that "Ih contribute[s] to low pass filtering properties of pyramidal cells", which first of all may not be technically correct under normal conditions; second, seems to contradict the authors' own data (Fig. 7F; S2A); and third, inappropriately references a study that certainly does not make any such claim.

4. It is disturbing that the authors were able to find and remove significant parts of their original data that were supposedly included in error but went unnoticed, sometimes resulting in revoking their original conclusions (i.e. strong correlation of Ih and patient age). Taking this into account with the other inconsistencies and methodological concerns that are still present, it is challenging to come away from this paper with any strong insight about human neurons and how their intrinsic properties may contribute to brain level oscillations.

5. The unexplained variance in basic physiological properties continues to be a subject of concern. The RMP for L2/3 neurons spans from -60 to -80 mV; for L5 it's -80 to -55 mV. The authors acknowledge this may be either biological variation or an experimental confound, but do nothing further to try to resolve this. If the authors performed a set of similar experiments on mouse or rat cortical neurons in L2/3 or L5 under the conventional, highly stereotyped conditions of brain slice preparation, would they still see this level of variance? How much of the variance in this manuscript is experimenter-driven quality control in terms of slice preparation and patch-clamp prowess and how much is a result of other forces, like real biological variability and/or human brain tissue condition? Speaking of which, are there inclusion criteria for human brain samples? One can imagine that depending on specifics of the surgery and the patient, the tissue could be in very different states, potentially contributing to the health on individual neurons and their resting membrane potential. How do the authors deal with this?

In summary, is it not clear what the authors want the reader to conclude at the end of this manuscript. The experiments and analysis have all largely already been performed in human neurons, and the attempt to connect these cellular properties to oscillatory dynamics, as has already been done in rodents, is weak. It is thus hard to understand how the current work will have a substantial impact in the field.

Reviewers' comments

Reviewer #1:

The authors have done a nice job of being responsive and open to the reviewer concerns and addressing the major issues raised. I appreciate the willingness of the authors to acknowledge various flaws that were either directly called out or discovered on their own in the course of revision. The corrections and new experimental data collected were quite important and no doubt have strengthened the manuscript over the original submission. I acknowledge that quite a lot of new experimental and analysis work was completed for this revision. Given the substantial new material introduced, I have additional comments on those changes, and these new comments are intended to help further strengthen the manuscript. I believe the manuscript could be acceptable for publication in Nature Communications pending these additional relatively minor changes.

We thank the reviewer for their positive comments on our work.

Conducting additional experiments on L3c pyramidal neurons was a reasonable compromise to get at gradients of features for supragranular pyramidal neuron population. This was an acceptable choice given that targeting of rare L5 thick tufted neurons in the human cortex would be exceptionally challenging. The conclusions are limited by the lack of L5 ET neurons in the data set, but this caveat is reasonably well discussed and acknowledged, and I find that acceptable.

We again thank the reviewer for their positive comments and for recognizing the additional experiments involved in our first round of revisions. In addition, Reviewer #1 has recognized one of the key messages of this manuscript: there is a clear diversity amongst pyramidal cells in human cortical L5, and we have situated our conclusions within this context.

Regarding L2/3 vs L3c, I worry that this naming scheme may cause confusion for readers familiar with rodent work on L2/3, as there are in fact discrete layers 2 and 3 anatomically in the human cortex. The authors should consider whether this can be better solved in the presentation of the data.

We thank the reviewer for this insightful point. In our revised manuscript, we have changed the naming scheme from L2/3 to L2&3, and discussed this rationale in the Methods (**Lines 135-137**). We feel this subtle change highlights that L2 and L3 are distinct in the human cortex (rather than indistinguishable in the rodent cortex), while not making this new naming schema unnecessarily confusing for those coming from the rodent setting.

Line 483-484 should be corrected for grammar, e.g. "These results suggest that depth from pia appears to be a general organizing principle for sag current in the cortex for both excitatory and inhibitory neurons." I'm not so sure organizing principle is the correct way to pitch this observation. It is an interesting observation, and is the only real justification to keep this as a main figure. It is a little odd to include this where interneurons in L2/3 and L5 are called "putative" and no information on the diversity of interneuron types is readily available for these data, especially

given that the rest of the paper is focused on excitatory neurons across layer. To me this seems to be a supplemental figure at best, and still feels preliminary. The principle result of sag currents being larger in deeper layers relative to upper layers may be a real phenomenon, or it may be a result of interneuron subtype sampling bias in this small interneuron dataset. The authors should consider this issue and at least acknowledge the limitations more directly in the text.

We very much appreciate this comment on one of the new figures in our manuscript. In our revised manuscript, we have adjusted the text (**Lines 584-597**) to more conspicuously acknowledge the limitations of this preliminary exploration of putative interneurons reported here. We have also taken the reviewer's suggestion to make this a supplementary figure (**Fig. S7**).

Regarding Figure S4 (note from the authors: this is now Figure S8 in the resubmitted manuscript), are these two neuron morphologies shown at matched scale? Likely they are not. It is a peculiar non-pyramidal neuron morphology indeed.

The scale was indeed correct for these two neurons. However, the two neurons were previously imaged in different fashions, due to COVID-19 restrictions that prevented a full 3D reconstruction of the non-pyramidal cell, and we used what was available to us to reconstruct a 2D morphology. With these restrictions loosened over the past few months, we have been able to perform a proper 3D reconstruction of this non-pyramidal neuron that should resolve any concerns in this regard (now **Figure S8**).

Figure S7 (note from the authors: this is now Figure S1 in the resubmitted manuscript): It is striking that the input resistance ranges are completely overlapping between Krembil and Allen data for L2 and L3 pyr neurons, yet the distributions are nearly non overlapping for L5 neurons, which can't be fully accounted for by truncation status. I imagine this discrepancy can't be resolved in the present study, yet it is notable. Despite this fact, it is commendable that the authors utilized this publicly available data set to compare to their data set and attempt to corroborate several findings.

We thank this reviewer for acknowledging our sincere attempts to reconcile our data with analogous datasets from the Allen Institute and elsewhere.

We also were puzzled that the input resistance values from the Allen Institute public L5 pyramidal cells were even larger than those reported within our primary dataset. However, both ours and the Allen datasets showed considerably larger input resistances than the thick-tufted L5 cells reported in Beaulieu-Laroche et al. (Beaulieu-Laroche, Toloza et al. 2018), further suggesting that ours and the Allen L5 data are perhaps more likely to be IT versus ET projecting cells. However, further reconciling such differences, such as input resistance values, between each of these datasets will likely require more robust approaches for identifying the same cell type across datasets, such as Patch-seq or the use of viral tools.

The novel visualization in Fig 6 is interesting but provides more questions than discrete answers. The recent transcriptomics data strongly suggests that layer membership is not particularly

informative regarding cell type diversity in human cortex, as many discrete cell types do not obey strict laminar boundaries. It might be worth expounding a little more on the point that layer is not a good surrogate for discrete cell types.

We thank the reviewer for this observation. We agree that one of the major points of this analysis is that it further highlights that layer membership is not a particularly informative dimension, on its own, for cell grouping (a point that also corroborates recent transcriptomic evidence). In our revision we have discussed this specific point in new text (**Lines 567-570**) as well as in a paragraph in the Discussion beginning on **Line 625**.

Reviewer #2:

For this revised manuscript, the authors report five main findings from their study on human neocortical pyramidal cells: 1) a gradient of increasing intrinsic excitability and Ih from superficial to deeper layers; 2) Ih-dependent enhancement of neuronal gain at delta and theta frequencies for layer 5 cells; 3) subthreshold resonance found in pyramidal cells; 4) heterogeneity in membrane properties; and 5) a possible correlation of Ih with patient age.

Most of these results are not new, having been previously investigated quite deeply, as was mentioned in the preceding round of review. The contribution of Ih to membrane resonance and its potential link with neural oscillation has been known for decades (Hutcheon et al., 1996). The positive correlation between Ih and somatic depth of human layer 2/3 pyramidal cells as well as its contribution to frequency-dependent power were also recently described in detail (Kalmbach et al., 2018). Stronger Ih in thick tufted human layer 5 pyramidal cells compared with layer 2/3, consistent with many previous observations from rodents, has likewise been documented (Beaulieu-Laroche et al., 2018). A large proportion of remaining data in the current manuscript, such as action potential waveform analysis, is potentially interesting, but seems detached from the main direction of the study.

The authors emphasize two findings in particular as important: direct experimental demonstration of the consequence of Ih block on frequency-dependent gain in layer 2/3 and 5 pyramidal cells (which was suggested in the previous round of review) and the potential correlation between Ih and age. However, these results are both not surprising based on previous knowledge and inadequately supported by the evidence provided. Additionally, serious concerns persist regarding methodology, data curation, and interpretation; this accrues with inconsistencies between the current study and the existing literature, calling in to question what descriptive value remains.

We appreciate the detailed reading of our manuscript that yielded “five main findings” as assessed by the reviewer. However, we feel that this contextualization of our manuscript minimizes two of the most important findings of our manuscript: first, that there is immense diversity amongst pyramidal cells in human L5, and second, that the correspondence between a large sag current (and corresponding activity of the h-current) and subthreshold resonance is not nearly as direct

as has been suggested by previous studies, including those cited by the reviewer here. These minimizations appear to arise from oversights regarding the literature on the electrophysiological characterization of human neurons and the study of resonance, which we will detail in the following.

Additionally, upon re-reading the manuscript with these comments from Reviewer #2 in mind, we recognized that our title and abstract might have contributed to a misplaced emphasis in the importance of our main findings. Therefore, we have **reworked the abstract** in this most recent revision, as well as **retitled the manuscript** as “Diversity amongst human cortical pyramidal neurons revealed via their sag currents and frequency preferences.”

We have also **eliminated the discussion of the correlation between I_h and patient age from the manuscript** entirely to avoid this “side-argument” confounding the larger results put forth in this paper.

Finally, in our revised manuscript we have **explained our findings in light of those from other groups**, including a more in-depth comparison of subthreshold resonance, morphologies, and intrinsic features such as resting potentials. In addition, we have more thoroughly compared our physiological and morphological findings of L5 human pyramidal cells to those of other groups and utilized results from other methodologies such as single-nucleus transcriptomics.

Major points:

1. The large majority of cells are shown to be non-resonant regardless of their position, with fewer cells displaying very low resonance frequency typically around 1 Hz even though representative examples are taken from the extremes (Fig. 4C). In contrast, Kalmbach et al. report the majority of layer 2/3 cells to be at least modestly resonant, especially in but not limited to the deepest layer 3 cells (Fig. 4B & D of Kalmbach et al.), along with considerably higher resonance frequency compared with values in the current manuscript. While the possible cause of inconsistency between these data is not discussed, the authors address this lack of subthreshold resonance by noting that there are other known contributors to membrane resonance.

We agree with the reviewer that the overall fractions of resonant cells as well as their resonant frequencies in our analysis are lower than those reported in Kalmbach et al. (Kalmbach, Buchin et al. 2018). However, our data replicates the trend reported in Kalmbach et al. that there is enhanced resonance for deeper layer L3 relative to more superficial cortical layers.

In this light we now more clearly explain and contextualize our results regarding subthreshold resonance with those reported in Kalmbach et al. (Kalmbach, Buchin et al. 2018). on **Lines 460 to 468**. This includes a direct discussion of the experimental limitations impacting the observation of subthreshold resonance on **Lines 465-468**. These limitations include the possibility of inadvertent dendrite truncation which might affect the observation of subthreshold resonance. An additional explanation is that such differences might arise due to differences in slice preparation and external and internal solutions used during recordings (e.g., we did not use synaptic blockers in our ZAP recordings).

Lastly, a likely alternative explanation consistent with our recent modeling work (Rich, Moradi Chameh et al. 2020) is that we delivered ZAP current stimuli at our cells' RMP, which tended to be slightly more depolarized than those in Kalmbach et al. (which we suspect to arise from solution differences, like those referred to above). This RMP difference likely influences the observation of resonance, including the fact that we are less likely to observe resonance than in the relevant figures in Kalmbach et al. referred to by the reviewer. This theory is supported by a relevant supplemental figure from Kalmbach et al. illustrating that when ZAP is delivered at a -65 mV holding potential (near the RMP for our L2&3), numerous cells were not resonant (i.e., they displayed a resonant peak at or near 0Hz, figure reproduced below). We note that the finding that subthreshold resonance at higher frequencies is more likely to arise at more hyperpolarized voltages, a conclusion drawn from the synthesis of the results of Kalmbach et al. and this study, comports with predictions made in our recently published modeling work (Rich, Moradi Chameh et al. 2020).

Quoting from **Lines 460-468**: “These results are generally consistent with recent evidence for greater subthreshold resonance in the deeper part of the supragranular layers of the human neocortex relative to more superficial neurons (Kalmbach, Buchin et al. 2018). While we observed a smaller fraction of resonant cells than previous work, we note that our results correspond with the conclusion that human L2&3 pyramidal cells are most likely to have normalized impedance peaks at <2 Hz, while neurons with peaks at >4 Hz are quite rare. Possible explanations for the lower fraction of resonant cells in our data include our use of different experimental solutions than Kalmbach et al., as well as the possibility of inadvertent dendrite truncation in these experiments (see Discussion). Additionally, our neurons displayed a slightly more depolarized RMP (Kalmbach, Buchin et al. 2018), which is a determinant of observing resonance.”

Unlike Kalmbach et al., such small fraction of resonant cells with very low resonance frequencies do not appear to be in good alignment with, or at least immediately relevant to, one of the central arguments of the manuscript, that I_h shapes neuronal resonance properties in such way that results in enhanced input-output relationship at delta-theta range for layer 5 cells (Fig. 7F) (note from the authors: this is now Figure 5f in the resubmitted manuscript). It could be argued that the difference may originate from the subthreshold vs. suprathreshold nature of the stimulation

protocols used; even so, the frequency-dependent gain profile from noisy current input (Fig. 7B) (note from the authors: this is now Figure 5b in the resubmitted manuscript), is not consistent with that from sinusoidal current input either (Fig. 4F), where the significant and roughly twofold difference in spike probability at higher frequencies as shown in the latter disappears from the former. More apparent discrepancies can be found from Fig. 7 (note from the authors: this is now Figure 5 in the resubmitted manuscript), wherein the frequency-dependent gain profiles in panels B vs. E & F show striking difference in tendency which in some ways even appear to be reversed, particularly for layer 5 cells.

In the revised manuscript, we have added text in multiple locations to better explain the perceived inconsistencies between our subthreshold resonance, suprathreshold resonance, and frequency dependent gain results. We feel that the reviewer's comment potentially represents a misconception regarding the relationship between these three distinct measures and I_h . We will expand upon this issue in more detail than allowed by the length limitations of the manuscript below.

It is first worth clarifying that we make no claim that " I_h shapes neural **resonance** properties in such way that results in enhanced input-output relationship at delta-theta range for layer 5 cells" (emphasis added). It remains an open question whether there is a "one-to-one" relationship between subthreshold resonance and suprathreshold frequency preference, as implied by this comment: in fact, computational and mathematical research (see new citations and text at **Lines 473-475**) has directly shown that the impact of subthreshold frequency preference on suprathreshold frequency preference is at best unclear. Our recent modelling work (Rich, Moradi Chameh et al. 2020) also supports the notion that resonance is a complex interplay between intrinsic membrane conductances, passive membrane properties, and other neural features, that makes this story more complex. We feel this point may have been further confounded by a misunderstanding by the reviewer regarding frequency dependent gain and suprathreshold resonance: namely, frequency dependent gain is a *distinct* measure from either sub- or suprathreshold resonance. While the analysis of a suprathreshold ZAP current via impedance plots measures the likelihood of a spike occurring from a sinusoidal input at a specific frequency, the frequency dependent gain measures the likelihood of a neuron spiking in phase with an oscillatory input that is relatively small (Yu and Lewis 1989, Higgs and Spain 2009). *These two measures provide complimentary, rather than superimposable, findings.*

We hope that this explanation helps to clarify why we feel that these points brought up by the reviewer do not represent inconsistencies. In an endeavor to make these nuances clearer to the reader, we have made the following changes to the manuscript:

- A revised statement of our main hypothesis on **Lines 75-77** clarifying that we propose that different biophysical and active properties (including, but not limited to, I_h) between superficial and deep layer human neurons underlie the dominant role of deep layer human neurons in driving interlaminar coherence: "*Based on our previous findings that deep layer activity appears to drive superficial activity in the human cortex (Florez, McGinn et al. 2015), we hypothesized that this "leading" role in generating interlaminar coherence*

can be attributed in part to the differing intrinsic properties of deep layer from superficial layer neurons.”

- An explicit mention of the distinct nature of frequency dependent gain analysis relative to resonance analysis in the Methods at **Lines 225-227**: *“This measure identifies the likelihood of the neuron spiking in phase with an oscillatory input that is small relative to the overall input to the cell, distinct from analysis of the neuron’s activity in response to a suprathreshold ZAP input (Higgs and Spain 2009).”*
- A more detailed comparison between the conclusions that can be drawn from both resonance and frequency dependent gain analysis, and their distinctness, before the presentation of the frequency dependent gain results in the Results at **Lines 497-502**: *“This measure captures distinct neuronal features compared to sub- or suprathreshold resonance: while resonance identifies the likelihood of a spike occurring from a drive at a particular frequency that is itself is suprathreshold, the frequency dependent gain quantifies the phase preference of neuronal spiking as a function of frequency (Yu and Lewis 1989) from a noisy input that is relatively small (Higgs and Spain 2009). Neurons with a high gain at a specific frequency are more likely to have a phase preference at that frequency than at other frequencies.”* Note that firing “in phase” with a sinusoid for frequency-dependent gain is distinct from a spike “occurring from a sinusoidal input at a specific frequency” as mentioned above for resonance.
- A reminder of this point in the discussion as we discuss the “dynamic circuit motif” at **Lines 662-664**: *“With subthreshold resonance not observed as a general feature of L5 pyramidal cells, we sought other biophysical features that might explain why L5 cells appear to drive interlaminar theta coherence.”*
- A final note regarding the interplay between I_h , resonance, and frequency dependent gain concluding this part of the Discussion on **Lines 689-692**: *“Interpreted together, our frequency dependent gain and ZAP results suggest that I_h may not be a direct “cause” of cortical oscillations at theta (~8Hz), but rather tune RS cells to follow with great fidelity the IB output at delta (see Fig. S10 for this dynamic circuit motif).”*

It is surprising in this regard that, despite such large variances in data, that the authors felt showing $n = 3$ cells without any statistical analysis was appropriate. I don’t even see any error bars. I do not find this experiment convincing, particularly because it’s not clear how these few cells were chosen from the heterogenous population shown in Fig 4.

In our initial revisions, we were specifically asked by both Reviewers 2 and 3 to use ZD to test whether peaks in frequency dependent gain were indeed driven by I_h via specific blockade. While of course we would like to increase our “ n ”, doing so is essentially impossible given current COVID-19 restrictions. In addition, applying ZD is a time consuming process that eliminates our ability to perform any additional experiments on a given slice, which given the limited access to human tissue is a consideration that must be taken into account when designing our experiments. Given these challenges, we feel that $n=3$ cells is acceptable.

In our revised manuscript, we have addressed the reviewer's suggestion to add error bars to this analysis (see **Figure 5e-f**, where the shaded area represents mean \pm one standard deviation). We have also indicated the resting membrane potential and input resistances of these three cells in the figure legend, to provide further context for how these cells fit into the broader landscape of cells that we have reported here.

2. It is remarkable that such cells as cell f or cell g shown in Fig. 1, which only reach up to layer 3 or even the border between layer 3 and 4, are presented as representative examples of layer 5 pyramidal cells that are presumed to be intact along the longitudinal axis. Cell g is further proposed to be a thick tufted layer 5 pyramidal cell (line 539), even though its morphological reconstruction simply lacks a tuft. Such features are completely unencountered in rodents and, to our knowledge, in humans as well. Note that the spiny neurons that are labeled as intact in the Allen Institute database whose electrophysiology and morphology are available (of which there are only two cells) also have visible apical dendrites reaching all the way up to layer 1, while at the same time displaying classic morphological features of neocortical layer 5 pyramidal cells.

We thank the reviewer for their careful review of the morphologies shown in Fig 1, and specifically for the comment that as cell g has no visible tuft, it should not be referred to as a putative thick-tufted cell - we have corrected this in the text on **Line 310-327** in our revised manuscript.

Below we have summarized previous reports of morphologies of human L5 pyramidal cells. In addition to the reconstructions reported in our study, there are 2 major published reports of human L5 pyramidal cell morphologies:

- Mohan et al., 2015, *Cerebral Cortex*: In this report, very few L5 pyramidal cells had dendrites reaching past L3, consistent with our study. The portion of their Figure 2 showing these L5 cells is included below for reference. As reported in this work: "To include a neuron for reconstruction, first the biocytin signal had to be dense and uniform throughout distal dendrites. Second, dendritic structures had to show minor cutting artifacts by the slicing procedure and had to be retained in the slice as much as can be expected realistically. These 2 criteria rejected about 80% of recovered neurons from reconstruction, mainly because of obvious truncation of the apical dendrite (Mohan, Verhoog et al. 2015)."

Figure 2. Dendrite gallery of human temporal cortex neurons. Representation of 91 3D reconstructed apical and basal dendrites of human temporal cortex (Brodmann area 21) arranged along somatic depth with respect to pial surface. Capitals (T, C, M) indicate that tissue was obtained from patients with subcortical tumor, cavernoma, or meningitis. First row indicates depth in μm . Apical dendrite in blue, basal dendrite in red. Neurons not labeled originate from patients with mesiotemporal sclerosis (MTS).

- Beaulieu-Laroche et al., 2018, *Cell*: According to the authors of this work, each of the pyramidal cells recorded had "thick apical dendrites reaching L1". In addition, they "targeted the thickest dendrites and biggest L5 somas to isolate putative L5B neurons"

(Hattox and Nelson 2007, Hay, Hill et al. 2011, Harnett, Xu et al. 2013, Harnett, Magee et al. 2015, Beaulieu-Laroche, Toloza et al. 2018).” Thus, we reason that these cells are likely to be extra-telencephalic and might not be immediately consistent with ours or those of other reports which did not specifically target recordings to the largest cells in L5. We have emphasized the distinction between ET and IT pyramidal cells in a multitude of locations in the revised manuscript, including in the Discussion (**Lines 649-652, 671-673, 630-639**) and various parts of the Results (**for example, Lines 536-543, 563-567**).

- Unpublished data of 2 L5 cells from Allen Institute: We agree with the reviewer that their 2 spiny L5 cells labeled with intact dendrites do indeed reach L1.

From these reports, and in particular the study of Mohan et al., we feel that the evidence is mixed whether all human L5 pyramidal cells have apical dendrites that reach L1. However, given the challenge of potential dendrite truncation when preparing slices containing such large cells, we cannot rule out that our representative L5 morphologies in Fig 1 have not been inadvertently truncated.

Lastly, we feel that the evidence is further mixed whether all mouse L5 pyramidal cells also have apical dendrites that reach L1. As one example, as reported recently in Adkins et al., (Adkins, Aldridge et al. 2020) and further illustrated in Scala et al., (Scala, Kobak et al. 2020) using the Patch-seq method which enables transcriptomically identifying IT-projecting L5 pyramidal cells and further reconstructing cell morphologies, in this study they report a number of IT-defined L5 pyramidal cells with apical dendrites that do not terminate in L1. See Figure 4 c, g reproduced below from this work illustrating these findings.

In the revised manuscript, we have summarized our findings and those of others in **Lines 321-327**. *“While we note that our human L5 morphologies are different from those reported by Beaulieu-Laroche et al. (Beaulieu-Laroche, Toloza et al. 2018) that targeted rare thick-tufted L5 pyramidal cells (Hodge, Bakken et al. 2019) with tufts reaching into L1 (Ramaswamy and Markram 2015), our cell morphologies are consistent with other previous reports that relatively few L5 neurons have dendrites extending past L3 (Mohan, Verhoog et al. 2015). Additionally, given the challenge of potential dendrite truncation when preparing slices containing such large cells (Mohan, Verhoog et al. 2015), it is possible that our representative L5 morphologies in Figure 1 have been inadvertently truncated (Mohan, Verhoog et al. 2015). However, we only observed visible truncation in one branch of one cell (the largest cell, cell g) and no obvious truncation in the other cells shown in Figure 1”.*

In addition, it is worthwhile to note here that the presence of spines alone is not a guarantee for a cell to be pyramidal or even excitatory, and interneurons including the sparsely spiny interneurons are also found more frequently in primates than in rodents (Kawaguchi & Kubota, 1993; DeFelipe, 2011). Whether the apical dendrites of cells recorded are intact is of much importance for the current study, not only as it will affect passive membrane properties but more importantly because ion channels including HCN channels are known to be expressed in steep gradients along the somatodendritic axis (Lorincz et al., 2002).

We agree that the presence or absence of apical dendrites is important for our study, especially as it relates to the somatodendritic distribution of HCN channels. In our revised manuscript, we now discuss at **Lines 706-712** how inadvertent dendrite truncation might impact properties that are thought to arise from I_h .

It is also unclear from the manuscript how the authors excluded the non-fast-spiking interneurons (which coincidentally have high input resistance) from analysis, which are much less obviously distinguishable from pyramidal cells by electrophysiology alone to the untrained eye. The authors describe that putative interneurons were identified by spike properties such as action potential width, maximum firing rate, and strong afterhyperpolarization, all of which are well-suited criteria for fast-spiking interneurons such as those shown as representative examples, but much less effective for identifying non-fast-spiking interneurons (a population which also overlaps with the aforementioned sparsely spiny interneurons such as calbindin-expressing interneurons in the deeper layers of the neocortex).

Following the comments of this reviewer and others, in our prior revision we undertook a systematic effort to carefully review electrophysiological data from each of the recorded neurons in our prior analysis. In particular, in this re-analysis we were able to identify and remove from our primary analysis a number of putative interneurons, i.e. those with characteristics reminiscent of more narrow-spiking interneurons. Using similar approaches, Torres-Gomez et al. showed that intracellular spike waveform features can sufficiently distinguish PV and SST cells from pyramidal cells (see Fig 5 in Torres-Gomez et al.). (Torres-Gomez, Blonde et al. 2020)

However, as the reviewer states (and further supported by the work in Torres-Gomez et al.), it is not possible to use intracellular electrophysiological features alone to further separate interneurons with more broad-spiking spiking features, such as VIP interneurons, from pyramidal cells. We acknowledge this as a limitation of our analysis and note this would be a limitation in any analysis where pyramidal cell identities were not confirmed post-hoc using morphological reconstructions or another methodology. However, VIP (and other CGE-derived) interneurons are relatively infrequent, making up approximately 5% of all neurons in L5 of the human neocortex. In the **new Fig S9** we have replotted cell type proportions based on transcriptomics data from MTG from Hodge et al. (Hodge, Bakken et al. 2019). Therefore, if VIP and other CGE-derived cells are present in our dataset, it is reasonable to expect that their prevalence would be somewhat rare. In our revised manuscript, we have explicitly addressed these issues in the Methods at **Lines 166-172**.

3. The authors continue to place irrelevant and inappropriate references in attempts to support their otherwise unfounded claims, even after having been corrected for numerous such instances from their previous manuscript. For example, they state that “human neocortical circuits demonstrate [...] different short-term plasticity rules, compared to neocortical circuits in rodents”; setting aside the ambiguity on exactly what type of synapses the authors are referring to here, one of the two cited papers is entirely on rhythmic firing in rat layer 5 pyramidal cells, while the other is about timing rules for spike timing dependent plasticity in human hippocampal (not neocortical) neurons. Neither study addresses or even mentions short-term plasticity. Another example is their statement that “Ih contribute[s] to low pass filtering properties of pyramidal cells”, which first of all may not be technically correct under normal conditions; second, seems to contradict the authors’ own data (Fig. 7F; S2A); and third, inappropriately references a study that certainly does not make any such claim.

While we took care to address this issue previously (and thank the reviewer for pointing out specific instances in their prior review), we apologize that inappropriate citations persisted after our initial revisions. We thank the reviewer for pointing out these 3 potentially erroneous citations. **We have corrected these specific citations and the corresponding text**, and also have rigorously gone through the remaining citations in an effort to ensure that no additional errors of this kind persist in our revised manuscript.

4. It is disturbing that the authors were able to find and remove significant parts of their original data that were supposedly included in error but went unnoticed, sometimes resulting in revoking their original conclusions (i.e. strong correlation of Ih and patient age). Taking this into account with the other inconsistencies and methodological concerns that are still present, it is challenging to come away from this paper with any strong insight about human neurons and how their intrinsic properties may contribute to brain level oscillations.

We again thank the reviewer for their comments in our first round of review, alerting us to potential data curation issues in our initial dataset. Upon receiving these comments from this and the other reviewers, we made our best attempts to systematically review our data and reconcile our findings with the past literature.

While we understand and appreciate the reviewer's concerns regarding this reanalysis, we maintain that this serves to strengthen the quality of our dataset and the resulting conclusions. We note that this opinion appears to be shared by Reviewer #1, as quoted in the most recent round of review: "The authors have done a nice job of being responsive and open to the reviewer concerns and addressing the major issues raised. ***I appreciate the willingness of the authors to acknowledge various flaws that were either directly called out or discovered on their own in the course of revision. The corrections and new experimental data collected were quite important and no doubt have strengthened the manuscript over the original submission. I acknowledge that quite a lot of new experimental and analysis work was completed for this revision***" (emphasis added).

We emphasize that our initial argument regarding patient age was a relatively minor part of our initial submission. As such, we have elected to remove this analysis entirely in the revised manuscript. We do not feel that this considerably affects the quality or novelty of our manuscript.

We feel that the "other inconsistencies and methodological concerns" have been addressed (see our response to Main Points 2 and 5), which should mitigate any concerns about the quality of our experimental data.

5. The unexplained variance in basic physiological properties continues to be a subject of concern. The RMP for L2/3 neurons spans from -60 to -80 mV; for L5 it's -80 to -55 mV. The authors acknowledge this may be either biological variation or an experimental confound but do nothing further to try to resolve this. If the authors performed a set of similar experiments on mouse or rat cortical neurons in L2/3 or L5 under the conventional, highly stereotyped conditions of brain slice preparation, would they still see this level of variance? How much of the variance in this manuscript is experimenter-driven quality control in terms of slice preparation and patch-clamp prowess and how much is a result of other forces, like real biological variability and/or human brain tissue condition?

We do not agree with the reviewer that the ranges of RMP are inconsistent with what one would expect. In fact, there are numerous examples in the human literature that correspond with the level of variability seen in our study, which we highlight below:

- For comparison: in Ting et al. 2018, *Scientific Reports*, an RMP range between approximately -60 and -80 mV is reported in L2 and L3 human pyramidal cells (see figure below, the relevant portion of Figure 3 from that work) (Ting, Kalmbach et al. 2018).

- In the work of Kalmbach et al., 2018, *Neuron* (see relevant portion of their Figure 2 below), similar variability is seen in the RMP amongst the human L2 and L3 neurons. (Kalmbach, Buchin et al. 2018).

- Similar variability in the RMP is seen amongst the human L2 and L3 neurons characterized by (Berg, Sorensen et al. 2020). (see figure below, the relevant portion of extend data Figure 5 from that work).

- While the work of (Beaulieu-Laroche, Toloza et al. 2018) shows less variability in RMP (approximately -70 to -55 mV, see relevant portion of Figure S1 taken from that work below), one possible explanation is that as they targeted the largest L5 pyramidal cells, these cells might reflect a more homogeneous population than the set of L5 cells targeted here (**a point that, as noted above, we have emphasized in our revised manuscript**).

On this point, it is worth noting that recordings from the same work in L2&3 (included as Supplemental Figure 2, the relevant portion of which is included below) shows a slightly increased range of RMPs, which is similar to the variability reported in our work.

Speaking of which, are there inclusion criteria for human brain samples? One can imagine that depending on specifics of the surgery and the patient, the tissue could be in very different states, potentially contributing to the health on individual neurons and their resting membrane potential. How do the authors deal with this?

Our inclusion criteria for human brain samples is provided in the Methods section on **Lines 94-96** in the revised manuscript. On general issues related to the quality of human surgical tissue, we point the reviewer to **Lines 713-724** of the revised manuscript. In short, we have taken care to acknowledge the challenges and caveats of making recordings from human surgical tissue, where tissue quality cannot be as precisely controlled as with rodent recordings.

Additionally we note that the existing human work by (Mohan, Verhoog et al. 2015, Kalmbach, Buchin et al. 2018) both delve into this topic in great detail, noting the necessary conventions used and accommodations afforded in the study of human neurons, given our limited access to human tissue and the resulting inability to “control” for as many factors as can be done in analogous rodent studies (see specifically the section entitled “Caveats of Human Tissue” in the Discussion of the Mohan et al. work). We explicitly cite these works to make this point on **Line 718** of the revised manuscript.

In general, as illustrated above for RMP values and in our prior revision for input resistance values (see Fig S1), we feel that our data are largely consistent with those reported previously. We thank this reviewer for encouraging us to perform this comparison as systematically and rigorously as possible. Following these comparisons, we now feel confident that unavoidable issues, such as surgical tissue quality, do not significantly compromise the results reported here.

In summary, is it not clear what the authors want the reader to conclude at the end of this manuscript. The experiments and analysis have all largely already been performed in human neurons, and the attempt to connect these cellular properties to oscillatory dynamics, as has already been done in rodents, is weak. It is thus hard to understand how the current work will have a substantial impact in the field.

Premised on the hypothesized functional differences in superficial and deep layer neurons as evidenced by collective neuronal dynamics, this work clearly addresses what we set out in our introduction: to explore the differences between human deep and superficial layer neurons in pursuit of a well-supported hypothesis explaining their functional differences. Through our explorations we show a large number of unique findings, and utilize complementary analysis to disentangle the functional implications of the differences between superficial and deep layer human neurons not previously done in human or rodent work.

We thus believe our work is impactful, a perspective we note is shared by Reviewer 1 and the original Reviewer 3, especially considering the posited relevance of using human inspired data to build human inspired models of the brain. We further note that, outside the subjective arguments of “impact”, *we can objectively state that our manuscript adds a valuable missing perspective on human L5 presumed-IT cells*. According to recent transcriptomic evidence (Hodge, Bakken et al. 2019), IT cells are expected to be far more prevalent in human L5 than the ET/thick-tufted cells that have been recently studied in (Beaulieu-Laroche, Toloza et al. 2018). This is included in our manuscript as the new **Figure S9**. We also feel that, given the rare opportunity to perform such experiments in live human tissue, it is extremely valuable to compare and contrast

our findings with those reported by other groups— *this is a strength of our manuscript, not a weakness.*

In conclusion, *we do believe our work will have substantial impact in the field, not only by improving our understanding of the intrinsic properties of human neurons, but also in providing one of the first explorations of the impact these properties may have on the oscillatory properties of the human neocortex.*

Original Reviewer #3 (and responses from the initial round of revisions):

This is a thorough, careful and thoughtful MS quantifying the distribution of Ih in human – mainly anterior temporal – neocortex by layer. I completely concur with the ethos behind the study – that modelling the human brain demands constraint by human data and, for this reason alone, ***the paper constitutes a valuable reference work and so to me, deserves publication.***

I do have a couple of issues with the MS as it stands though. ***The comments below are, hopefully, easily dealt with and are intended merely as suggestions to improve the MS.***

We have added emphasis to two key points from the original Reviewer #3's comments, which we feel speak to value of our manuscript (and perhaps might help address some of these concerns raised by Reviewer #2).

1) I see why the authors 'hang' their findings on the human theta rhythm. But I don't think its valid given precedents in rodent literature. As they argue in the discussion, there are multiple mechanisms that can underlie neuronal rhythmicity at theta frequency. Ih is almost certainly not one of them in neocortex. Hippocampal theta cannot reliably be considered the same phenomenon, the frequencies are different for a start. Here there is a dependence on Ih but it lies principally in the behaviour of a subset of interneurons and the neocortical equivalents were not examined in the present MS. In addition, Ih is exquisitely dependent on neuromodulatory state: Theta is mainly seen in non-invasive human recordings in the wake state but, in this condition, multiple wake-associated neuromodulators all act to reduce Ih considerably (e.g. Ach, Orexin/hypocretin etc.). What Ih IS critically involved in is rebound following synaptic inhibition. Thus it plays a crucial role in delayed responses to sensory input (the 'off' response) and 'anodal break' spiking seen in mismatch responses.

We thank the reviewer for this thorough and positive comment. We indeed agree with the reviewer's analysis that theta rhythmicity is a complex dynamic, both at the cellular and network level, that is driven by a wide variety of factors beyond I_h , and that there are important differences between human and rodent theta rhythms. We have thus endeavored to downplay the emphasis on theta resonance. Our ZD-7288 (I_h blocker) experiments firmly establish that the low frequency peaks in $G(f)$ particularly in the delta frequency are dependent on I_h . In light of these new results, we interpret our previous findings of interlaminar coherence at theta frequency (4-8Hz), as arising from the dynamic interaction between IB cells (that burst at delta), and RS cells (which are the predominant cell-type we likely recorded from) that receive delta frequency input from IB cells to which they are particularly tuned to (peak in delay in $G(f)$), which then discharge at theta frequency

(and possibly in part the theta frequency peak in $G(f)$). Please see response to last comment for a continuation of this discussion (**Figure S10 in revised manuscript**).

2) The data is presented with commendable clarity, but consequently suggests multimodal distributions in a number of the intrinsic cell properties measured. This is discussed briefly (L2 vs L3 for example) but the main subdivision of cell type I know of that manifests in part as sag amplitude differences is between L5a and L5b. The authors have gathered an impressive set of data so I wonder if it is possible to stratify L2/3 and L5 cell types further on the basis of other intrinsic properties (slow AHP, burst generation, after depolarisation strength etc.) This may help to 'clean up' the often very broad distributions in some of the metrics.

We thank the reviewer for this excellent comment. We point the reviewer to our analysis in what is now Fig 6, which attempts to address some of the large degree in heterogeneity within cells from the same cortical layer.

We also note that we address this topic in our responses to similar comments from the other reviewers: specifically, we have addressed some of this increased variability by identifying interneurons and distinct experimental protocols that were mistakenly included in our data set, while noting that some level of increased heterogeneity in these cells (relative to the rodent setting) is to be expected based on the emerging human cortical cell typing literature.

2 minor points:

3) Lack of observed resonance on somatic recordings with patch electrodes is not surprising. Patch solutions dialyse cytosol hugely and thus interfere with many intrinsic conductances. In addition, the distribution of I_h in neurons is not uniform and resonance can be seen in dendritic recordings in a given cell type when it appears completely absent in somatic recording.

We thank the reviewer for noting these potentials confounds in our subthreshold resonance data. We have now added a paragraph to our discussion within our limitations section to contextualize how experimental confounds, such as dendrite cutting or cytosol dialysis, might influence some of our ability to observe subthreshold resonance in these data.

4) The MS data does agree with the Carracedo data in terms of layers showing most theta (discussion). In that paper the field theta was mainly manifest in superficial layers but the origin of the synaptic activity underlying this was exclusively intrinsic theta activity in a subpopulation of L5 cells. See point 1 above though, this theta was not I_h -dependent.

We thank the reviewer for this excellent comment and insights. We fully agree that I_h is likely not directly responsible for the theta frequency activity in local circuitry. We do however argue in line with the (Carracedo, Kjeldsen et al. 2013) work, that it is indirectly complicit in theta generation given our new experiments added to this MS, that show that the delta peak (~2Hz) in $G(f)$, and less so the theta peak (~7Hz) are dependent on I_h (see Figure 5e-f). From the (Carracedo, Kjeldsen et al. 2013) paper it was the RS cells that generated theta, likely the cell-type that primarily contributed to $G(f)$ (Figure 5b). We speculate that they did not observe theta in the deep

layers, since human circuits appear to amplify local activity relative to rodent cortex (Molnár, Oláh et al. 2008) which might make theta more prominent in human L5. Conversely, we did not observe delta activity in our previous in-vitro recordings, possibly due to the paucity of IB cells in human middle temporal gyrus cortex (Hodge, Bakken et al. 2019). Our data as well provides a putative cellular mechanism (peaks in G(f)) underlying these population activities. We have schematized these ideas below and include it as a supplementary **Figure (S10)** and clarified this in the discussion.

References:

- Adkins, R. S., A. I. Aldridge, S. Allen, S. A. Ament, X. An, E. Armand, G. A. Ascoli, T. E. Bakken, A. Bandrowski and S. Banerjee (2020). "A multimodal cell census and atlas of the mammalian primary motor cortex." bioRxiv.
- Beaulieu-Laroche, L., E. H. Toloza, M.-S. van der Goes, M. Lafourcade, D. Barnagian, Z. M. Williams, E. N. Eskandar, M. P. Frosch, S. S. Cash and M. T. Harnett (2018). "Enhanced dendritic compartmentalization in human cortical neurons." Cell **175**(3): 643-651. e614.
- Berg, J., S. A. Sorensen, J. T. Ting, J. A. Miller, T. Chartrand, A. Buchin, T. E. Bakken, A. Budzillo, N. Dee and S.-L. Ding (2020). "Human cortical expansion involves diversification and specialization of supragranular intratelencephalic-projecting neurons." BioRxiv.
- Carracedo, L. M., H. Kjeldsen, L. Cunnington, A. Jenkins, I. Schofield, M. O. Cunningham, C. H. Davies, R. D. Traub and M. A. Whittington (2013). "A neocortical delta rhythm facilitates reciprocal interlaminar interactions via nested theta rhythms." Journal of Neuroscience **33**(26): 10750-10761.
- Florez, C., R. McGinn, V. Lukankin, I. Marwa, S. Sugumar, J. Dian, L.-N. Hazrati, P. Carlen, L. Zhang and T. Valiante (2015). "In vitro recordings of human neocortical oscillations." Cerebral Cortex **25**(3): 578-597.
- Harnett, M. T., J. C. Magee and S. R. Williams (2015). "Distribution and function of HCN channels in the apical dendritic tuft of neocortical pyramidal neurons." Journal of Neuroscience **35**(3): 1024-1037.
- Harnett, M. T., N.-L. Xu, J. C. Magee and S. R. Williams (2013). "Potassium channels control the interaction between active dendritic integration compartments in layer 5 cortical pyramidal neurons." Neuron **79**(3): 516-529.
- Hattox, A. M. and S. B. Nelson (2007). "Layer V neurons in mouse cortex projecting to different targets have distinct physiological properties." Journal of neurophysiology **98**(6): 3330-3340.
- Hay, E., S. Hill, F. Schürmann, H. Markram and I. Segev (2011). "Models of neocortical layer 5b pyramidal cells capturing a wide range of dendritic and perisomatic active properties." PLoS Comput Biol **7**(7): e1002107.
- Higgs, M. H. and W. J. Spain (2009). "Conditional bursting enhances resonant firing in neocortical layer 2–3 pyramidal neurons." Journal of Neuroscience **29**(5): 1285-1299.
- Hodge, R. D., T. E. Bakken, J. A. Miller, K. A. Smith, E. R. Barkan, L. T. Graybiel, J. L. Close, B. Long, N. Johansen and O. Penn (2019). "Conserved cell types with divergent features in human versus mouse cortex." Nature **573**(7772): 61-68.
- Kalmbach, B. E., A. Buchin, B. Long, J. Close, A. Nandi, J. A. Miller, T. E. Bakken, R. D. Hodge, P. Chong and R. de Frates (2018). "h-Channels Contribute to Divergent Intrinsic Membrane Properties of Supragranular Pyramidal Neurons in Human versus Mouse Cerebral Cortex." Neuron **100**(5): 1194-1208. e1195.
- Mohan, H., M. B. Verhoog, K. K. Doreswamy, G. Eyal, R. Aardse, B. N. Lodder, N. A. Goriounova, B. Asamoah, A. C. B. Brakspear and C. Groot (2015). "Dendritic and axonal architecture of individual pyramidal neurons across layers of adult human neocortex." Cerebral Cortex **25**(12): 4839-4853.
- Molnár, G., S. Oláh, G. Komlósi, M. Füle, J. Szabadics, C. Varga, P. Barzó and G. Tamás (2008). "Complex events initiated by individual spikes in the human cerebral cortex." PLoS biology **6**(9): e222.
- Ramaswamy, S. and H. Markram (2015). "Anatomy and physiology of the thick-tufted layer 5 pyramidal neuron." Frontiers in cellular neuroscience **9**: 233.
- Rich, S., H. Moradi Chameh, V. Sekulic, T. A. Valiante and F. K. Skinner (2020). "Modeling Reveals Human–Rodent Differences in H-Current Kinetics Influencing Resonance in Cortical Layer 5 Neurons." Cerebral Cortex.

Scala, F., D. Kobak, M. Bernabucci, Y. Bernaerts, C. R. Cadwell, J. R. Castro, L. Hartmanis, X. Jiang, S. R. Lathunus and E. Miranda (2020). "Phenotypic variation within and across transcriptomic cell types in mouse motor cortex." bioRxiv.

Ting, J. T., B. Kalmbach, P. Chong, R. de Frates, C. D. Keene, R. P. Gwinn, C. Cobbs, A. L. Ko, J. G. Ojemann and R. G. Ellenbogen (2018). "A robust ex vivo experimental platform for molecular-genetic dissection of adult human neocortical cell types and circuits." Scientific reports **8**(1): 1-13.

Torres-Gomez, S., J. D. Blonde, D. Mendoza-Halliday, E. Kuebler, M. Everest, X. J. Wang, W. Inoue, M. O. Poulter and J. Martinez-Trujillo (2020). "Changes in the Proportion of Inhibitory Interneuron Types from Sensory to Executive Areas of the Primate Neocortex: Implications for the Origins of Working Memory Representations." Cerebral Cortex.

Yu, X. and E. R. Lewis (1989). "Studies with spike initiators: linearization by noise allows continuous signal modulation in neural networks." IEEE Transactions on Biomedical Engineering **36**(1): 36-43.

Reviewer #1 (Remarks to the Author):

I am satisfied with the revisions by the authors. I recommend that the manuscript should be accepted in present form for publication in Nature Communications. The revised Figure material that was more appropriate for the supplement has been moved to the supplement, as requested, with additional changes to the corresponding text. The authors have made their best attempt to address concerns of all authors. The human slice physiology presented in this study is imperfect but is honestly represented and the authors do their due diligence in trying to reconcile their findings with other studies of similar scope. For example, the response about the wider than anticipated range of RMP values for human pyramidal neurons was clearly answered with compelling evidence that this is within the expected range for human cortical pyramidal neurons. The addition of figure S9 is really helpful to explain the sampling issue and how abundant L2/3 IT pyramidal neurons are in contrast to more rare L5 types, as well as how rare the undersampled L5 ET or thick-tufted neurons type is in human temporal cortex. This is a limitation of the approach of recording in unlabeled tissue slices and the authors have reasonably acknowledged the limitation. I also appreciated the points about intactness of L5 pyramidal neurons and absence of apical tufts. It was compelling and interesting to compare to examples from the largest study of human neuron reconstructions to date (Mohan et al) as well as to the mouse Patch-seq study from Tolias lab where even some mouse L5 IT neurons don't have clear apical tufts. It remains unresolved if L5 IT neurons in human (and mouse) cortex can exist without an apical tuft, if this feature varies by brain region or finer subtypes, and what is the functional significance of tufted vs non-tufted L5 neurons (although there are obvious implications). This will be a fascinating topic for future investigation.

I strongly believe that the novelty is an important factor, but there is ample room for incremental advances and more than just a handful of publications on physiology and morphology of human cortical pyramidal neurons. We need more work like this published, not less. Each study is informative to the field about how to make advances and increase the rigor of the work for future investigations in spite of the major challenges to doing human brain slice recordings.

Jonathan T. Ting